# Nonparametric Teaching for Graph Property Learners

Chen Zhang [1] [*]   Weixin Bu [2] [*]   Zeyi Ren [1]   Zhengwu Liu [1]   Yik-Chung Wu [1]   Ngai Wong [1]

## Abstract

Inferring properties of graph-structured data, *e.g.*, the solubility of molecules, essentially involves learning the implicit mapping from graphs to their properties. This learning process is often costly for graph property learners like Graph Convolutional Networks (GCNs). To address this, we propose a paradigm called **Gra**ph **N**eural **T**eaching (GraNT) that reinterprets the learning process through a novel nonparametric teaching perspective. Specifically, the latter offers a theoretical framework for teaching implicitly defined (*i.e.*, nonparametric) mappings via example selection. Such an implicit mapping is realized by a dense set of graph-property pairs, with the GraNT teacher selecting a subset of them to promote faster convergence in GCN training. By analytically examining the impact of graph structure on parameter-based gradient descent during training, and recasting the evolution of GCNs—shaped by parameter updates—through functional gradient descent in nonparametric teaching, we show *for the first time* that teaching graph property learners (*i.e.*, GCNs) is consistent with teaching structure-aware nonparametric learners. These new findings readily commit GraNT to enhancing learning efficiency of the graph property learner, showing significant reductions in training time for graph-level regression (-36.62%), graph-level classification (-38.19%), node-level regression (-30.97%) and node-level classification (-47.30%), all while maintaining its generalization performance.

## 1. Introduction

Graph-structured data, commonly referred to as graphs, are typically represented by vertices and edges (Hamilton et al.,

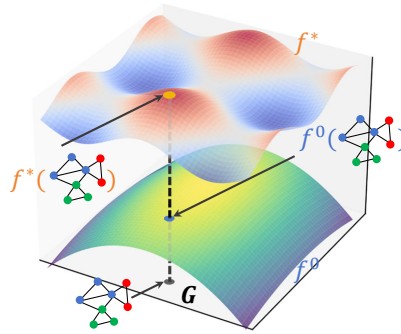

Figure 1: An illustration of the implicit mapping $f^*$ between a graph $G$ and its property $f^*(G)$, where $f^0$ denotes the mapping of the initial graph property learner, *e.g.*, an initialized GCN.

2017; Chami et al., 2022). The vertices, or nodes, contain individual features, while the edges link these nodes and capture the structural information, collectively forming a complete graph. Graph properties can be categorized as either node-level or graph-level[1]. For example, the node category is a node-level property in social network graphs (Fan et al., 2019), while the solubility of molecules is a graph-level property in molecular graphs (Ramakrishnan et al., 2014). Inferring these graph properties essentially involves learning the implicit mapping from graphs to these properties (Hamilton et al., 2017). An intuitive illustration of this mapping is provided in Figure 1. As a representative graph property learner, the Graph Convolutional Network (GCN) (Defferrard et al., 2016; Kipf & Welling, 2017) has shown strong generalizability, delivering impressive performance across various fields such as social networks (Min et al., 2021; Li et al., 2023), quantum chemistry (Gilmer et al., 2017; Mansimov et al., 2019), and biology (Stark et al., 2006; Burkhart et al., 2023).

However, the learning process of the implicit mapping—*i.e.*, the training—can be quite expensive for GCNs, particularly when dealing with large-scale graphs (Liu et al., 2022a). For example, learning node-level properties in real-world e-commerce relational networks involves millions of

---
[*]Equal contribution   [1]Department of Electrical and Electronic Engineering, The University of Hong Kong, HKSAR, China   [2]Reversible Inc.   Correspondence to: Chen Zhang <czhang6@connect.hku.hk>, Ngai Wong <nwong@eee.hku.hk>.

*Proceedings of the 42$^{nd}$ International Conference on Machine Learning*, Vancouver, Canada. PMLR 267, 2025. Copyright 2025 by the author(s).

---
[1]This paper adopts a graph-level focus in its discussion unless otherwise specified, with node-level considered as a multi-dimensional generalization.

Our project page is available at `https://chen2hang.github.io/_publications/nonparametric_teaching_for_graph_proerty_learners/grant.html`.

nodes (Robinson et al., 2024). In the case of graph-level property learning tasks, the scale can become prohibitively large (Hu et al., 2021). As a result, there is a pressing need to reduce training costs and improve the learning efficiency.

Recent studies on nonparametric teaching (Zhang et al., 2023b;a; 2024a) offer a promising solution to the above problem. Specifically, nonparametric teaching provides a theoretical framework for efficiently selecting examples when the target mapping (*i.e.*, either a function or a model) being taught is nonparametric, *i.e.*, implicitly defined. It builds on the idea of machine teaching (Zhu, 2015; Zhu et al., 2018)—involving designing a training set (dubbed the teaching set) to help the learner rapidly converge to the target functions—but relaxes the assumption of target functions being parametric (Liu et al., 2017; 2018), allowing for the teaching of nonparametric (viz. non-closed-form) target functions, with a focus on function space. Unfortunately, these studies focus solely on regular feature data and overlook the structural aspects of the inputs, resulting in difficulty when the inputs are irregular graphs—universal data structures that include both features and structure (Chami et al., 2022). Moreover, the update of a GCN is generally carried out through gradient descent in parameter space, leading to a gap compared to the functional gradient descent used in nonparametric teaching within function space (Zhang et al., 2023b;a; 2024a). These call for more examination prior to the adoption of nonparametric teaching theory in the context of graph property learning.

To this end, we systematically explore the impact of graph structure on GCN gradient-based training in both parameter and function spaces. Specifically, we analytically examine the impact of the adjacency matrix, which encodes the graph structure, on parameter-based gradient descent within parameter space, and explicitly show that the parameter gradient maintains the same form when the graph size is scaled. The structure-aware update in parameter space drives the evolution of GCN, which can be expressed using the dynamic graph neural tangent kernel (GNTK) (Du et al., 2019; Krishnagopal & Ruiz, 2023), and is then cast into function space. We prove that this dynamic GNTK converges to the structure-aware canonical kernel utilized in functional gradient descent, suggesting that the evolution of GCN under parameter gradient descent is consistent with that under functional gradient descent. Therefore, it is natural to interpret the learning process of graph properties via the theoretical lens of nonparametric teaching: the target mapping is realized by a dense set of graph-property pairs, and the teacher selects a subset of these pairs to provide to the GCN, ensuring rapid convergence of this graph property learner. Consequently, to enhance the learning efficiency of GCN, we introduce a novel paradigm called GraNT, where the teacher adopts a counterpart of the greedy teaching algorithm from nonparametric teaching for graph property

learning, specifically by selecting graphs with the largest discrepancy between their property true values and the GCN outputs. Lastly, we conduct extensive experiments to validate the effectiveness of GraNT in a range of scenarios, covering both graph-level and node-level tasks. Our key contributions are listed as follows:

- We propose GraNT, a novel paradigm that interprets graph property learning within the theoretical context of nonparametric teaching. This enables the use of greedy algorithms from the latter to effectively enhance the learning efficiency of the graph property learner, GCN.

- We analytically examine the impact of graph structure on parameter-based gradient descent within parameter space, and reveal the consistency between the evolution of GCN driven by parameter updates and that under functional gradient descent in nonparametric teaching. We further show that the dynamic GNTK, stemming from gradient descent on the parameters, converges to the structure-aware canonical kernel of functional gradient descent. These connect nonparametric teaching theory to graph property learning, thus expanding the applicability of nonparametric teaching in the context of graph property learning.

- We demonstrate the effectiveness of GraNT through extensive experiments in graph property learning, covering regression and classification at both graph and node levels. Specifically, GraNT saves training time for graph-level regression (-36.62%), graph-level classification (-38.19%), node-level regression (-30.97%) and node-level classification (-47.30%), while upkeeping its generalization performance.

## 2. Related Works

**Graph property learning**. Due to the versatility of graphs in modeling diverse data types (Chami et al., 2022), there has been a recent surge of research interest on graphs (Xia et al., 2021), especially attempts of learning implicit mapping from graph data to interested properties (Guo et al., 2021; Zhuang et al., 2023; Cao et al., 2023) for diverse downstream tasks, such as those related to proteins (Fout et al., 2017; Gligorijević et al., 2021) and molecular fingerprints (Duvenaud et al., 2015). There have been various efforts to the learner design for better mapping learning, such as the GCN learner (Defferrard et al., 2016; Kipf & Welling, 2017), which borrows the idea of convolutional neural networks used in image tasks (LeCun et al., 2015), and the graph attention network, which applies the attention operation (Veličković et al., 2018), and to the learning efficiency (Chen et al., 2018; Liu et al., 2022b; Zhang et al., 2023c), such as normalization (Cai et al., 2021), graph decomposition (Xue et al., 2023) and lazy update (Narayanan et al., 2022). Differently, we approach graph property

learning from a fresh perspective of nonparametric teaching (Zhang et al., 2023b;a) and adopt a corresponding version of the greedy algorithm to enhance the training efficiency of GCN.

**Nonparametric teaching**. Machine teaching (Zhu, 2015; Zhu et al., 2018) focuses on designing a teaching set that enables the learner to quickly converge to a target model function. It can be viewed as the reverse of machine learning: while machine learning seeks to learn a mapping from a given training set, machine teaching aims to construct the set based on a desired mapping. Its effectiveness has been demonstrated across various domains, including crowdsourcing (Singla et al., 2014; Zhou et al., 2018), robustness (Alfeld et al., 2017; Ma et al., 2019; Rakhsha et al., 2020), and computer vision (Wang et al., 2021; Wang & Vasconcelos, 2021). Nonparametric teaching (Zhang et al., 2023b;a) advances iterative machine teaching (Liu et al., 2017; 2018) by broadening the parameterized family of target mappings to include a more general nonparametric framework. In addition, the practical effectiveness of this theoretical framework has been confirmed in improving the efficiency of multilayer perceptrons (MLPs) when learning implicit mappings from signal coordinates to their corresponding values (Sitzmann et al., 2020; Tancik et al., 2020; Luo et al., 2023; 2024; 2025; Zhang et al., 2024a). Nevertheless, the limited focus on the structural aspects of the input in these studies makes it difficult to directly apply their findings to general tasks involving graph-structured data (Hamilton et al., 2017; Chami et al., 2022). This work systematically examines the impact of graph structure and highlights the alignment between the evolution of GCN driven by parameter updates and that guided by functional gradient descent in nonparametric teaching. These insights, for the first time, broaden the scope of nonparametric teaching theory in graph property learning and position our GraNT as a means to improve the learning efficiency of GCN.

## 3. Background

**Notation**.[2] Let $G_{(n)} = (\mathcal{V}, \mathcal{E})$ be a graph, where $\mathcal{V}$ denotes the set of $n$ vertices (nodes) and $\mathcal{E}$ denotes the set of edges. The $d$-dimensional feature vector is denoted as $[x_i]_d = (x_1, \cdots, x_d)^\top \in \mathbb{R}^d$, where the entries $x_i$ are indexed by $i \in \mathbb{N}_d$ ($\mathbb{N}_d := \{1, \cdots, d\}$). For simplicity, this feature vector may be denoted as $\boldsymbol{x}$. The collection of feature vectors for all nodes is represented by an $n \times d$ feature matrix $\boldsymbol{X}_{n \times d}$ (abbreviated as $\boldsymbol{X}$). The $i$-th row and $j$-th column of this matrix, corresponding to the $i$-th node and $j$-th feature, are denoted by $\boldsymbol{X}_{(i,:)}$ and $\boldsymbol{X}_{(:,j)}$, respectively. Equivalently, these can be expressed as $\boldsymbol{e}_i^\top \boldsymbol{X}$ and $\boldsymbol{X}\boldsymbol{e}_j$, where $\boldsymbol{e}_i$ is a basis vector with its $i$-th entry equal to 1 and all other entries equal to 0. The structure of the graph $G_{(n)}$

is captured by its adjacency matrix $\boldsymbol{A} \in \mathbb{R}^{n \times n}$, allowing the graph to be concisely represented as $\boldsymbol{G} = (\boldsymbol{X}, \boldsymbol{A}) \in \mathbb{G}$. The property of the graph is denoted by $\boldsymbol{y} \in \mathcal{Y}$, where $\boldsymbol{y}$ is a scalar for graph-level properties (*i.e.*, $\mathcal{Y} \subseteq \mathbb{R}$) and a vector for node-level properties (*i.e.*, $\mathcal{Y} \subseteq \mathbb{R}^n$). A set containing $m$ elements is written as $\{a_i\}_m$. If $\{a_i\}_m \subseteq \{a_i\}_n$ holds, then $\{a_i\}_m$ represents a subset of $\{a_i\}_n$ of size $m$, with indices $i \in \mathbb{N}_n$. A diagonal matrix with elements $a_1, \cdots, a_m$ is denoted by $\mathrm{diag}(a_1, \cdots, a_m)$, and if all $m$ entries are identical, it is simplified as $\mathrm{diag}(a; m)$.

Consider $K(\boldsymbol{G}, \boldsymbol{G}') : \mathbb{G} \times \mathbb{G} \mapsto \mathbb{R}$ as a symmetric and positive definite graph kernel (Vishwanathan et al., 2010). It can also be written as $K(\boldsymbol{G}, \boldsymbol{G}') = K_{\boldsymbol{G}}(\boldsymbol{G}') = K_{\boldsymbol{G}'}(\boldsymbol{G})$, and for simplicity, $K_{\boldsymbol{G}}(\cdot)$ can be abbreviated to $K_{\boldsymbol{G}}$. The reproducing kernel Hilbert space (RKHS) $\mathcal{H}$ associated with $K(\boldsymbol{G}, \boldsymbol{G}')$ is defined as the closure of the linear span $\{f : f(\cdot) = \sum_{i=1}^r a_i K(\boldsymbol{G}_i, \cdot), a_i \in \mathbb{R}, r \in \mathbb{N}, \boldsymbol{G}_i \in \mathbb{G}\}$, equipped with the inner product $\langle f, g \rangle_{\mathcal{H}} = \sum_{ij} a_i b_j K(\boldsymbol{G}_i, \boldsymbol{G}_j)$, where $g = \sum_j b_j K_{\boldsymbol{G}_j}$ (Liu & Wang, 2016; Zhang et al., 2023b;a). Rather than assuming the ideal case with a closed-form solution $f^*$, we consider the more practical scenario where the realization of $f^*$ is given (Zhang et al., 2023b;a; 2024a). For simplicity, we assume the function is scalar-valued, aligning with the focus on the graph level in this discussion[3]. Given the target mapping $f^* : \mathbb{G} \mapsto \mathcal{Y}$, it uniquely returns $\boldsymbol{y}_\dagger$ for the corresponding graph $\boldsymbol{G}_\dagger$ such that $\boldsymbol{y}_\dagger = f^*(\boldsymbol{G}_\dagger)$. With the Riesz–Fréchet representation theorem (Lax, 2002; Schölkopf et al., 2002; Zhang et al., 2023b), the evaluation functional is defined as follows:

**Definition 1.** *Let $\mathcal{H}$ be a reproducing kernel Hilbert space with a positive definite graph kernel $K_{\boldsymbol{G}} \in \mathcal{H}$, where $\boldsymbol{G} \in \mathbb{G}$. The evaluation functional $E_{\boldsymbol{G}}(\cdot) : \mathcal{H} \mapsto \mathbb{R}$ is defined with the reproducing property as follows:*

$$E_{\boldsymbol{G}}(f) = \langle f, K_{\boldsymbol{G}}(\cdot) \rangle_{\mathcal{H}} = f(\boldsymbol{G}), f \in \mathcal{H}. \tag{1}$$

Additionally, for a functional $F : \mathcal{H} \mapsto \mathbb{R}$, the Fréchet derivative (Coleman, 2012; Liu, 2017; Zhang et al., 2023b) of $F$ is given as follows:

**Definition 2.** *(Fréchet derivative in RKHS) The Fréchet derivative of a functional $F : \mathcal{H} \mapsto \mathbb{R}$ at $f \in \mathcal{H}$, denoted by $\nabla_f F(f)$, is implicitly defined by $F(f + \epsilon g) = F(f) + \langle \nabla_f F(f), \epsilon g \rangle_{\mathcal{H}} + o(\epsilon)$ for any $g \in \mathcal{H}$ and $\epsilon \in \mathbb{R}$. This derivative is also a function in $\mathcal{H}$.*

**Graph convolutional network (GCN)** is proposed to learn the implicit mapping between graphs and their properties (Kipf & Welling, 2017; Xu et al., 2018). Specifically, a $L$-layer GCN $f_\theta(\boldsymbol{G}) \equiv \boldsymbol{X}^{(L)}$ resembles a $L$-layer MLP,

---

[2]A notation table is provided in Appendix A.1.

[3]In nonparametric teaching, extending from scalar-valued functions to vector-valued ones, which pertains to node-level properties, is a well-established generalization in Zhang et al., 2023a.

with the key difference being the feature aggregation at the start of each layer, which is based on the adjacency matrix $\boldsymbol{A}$ (Wu et al., 2019; Krishnagopal & Ruiz, 2023).

$$\boldsymbol{X}^{(\ell)} = \sigma\left((\boldsymbol{A} + \boldsymbol{I})^{\kappa_\ell} \boldsymbol{X}^{(\ell-1)} \boldsymbol{W}^{(\ell)}\right), \ell \in \mathbb{N}_{L-1}$$

$$\boldsymbol{X}^{(L)} = \rho\left((\boldsymbol{A} + \boldsymbol{I})^{\kappa_L} \boldsymbol{X}^{(L-1)} \cdot \boldsymbol{W}^{(L)}\right), \quad (2)$$

where $\boldsymbol{W}^{(\ell)}$ is the weight matrix at layer $\ell$, with dimensions $h_{\ell-1} \times h_\ell$, $h_\ell$ denotes the width of layer $\ell$ with $h_0 = d$, and $\kappa_\ell$ denotes the convolutional order at the $\ell$-th layer. $\boldsymbol{X}^{(0)}$ is the input feature matrix, $\sigma$ represents activation function (*e.g.*, ReLU), and $\rho$ refers to the pooling operation (*e.g.*, $\rho(\boldsymbol{a}) = \mathbf{1}^\top \boldsymbol{a}$ for summation pooling).

**Nonparametric teaching** (Zhang et al., 2023b) is defined as a functional minimization over a teaching sequence $\mathcal{D} = \{(\boldsymbol{x}^1, y^1), \dots (\boldsymbol{x}^T, y^T)\}$, where the input $\boldsymbol{x} \in \mathbb{R}^d$ represents regular feature data without considering structure, with the set of all possible teaching sequences denoted as $\mathbb{D}$:

$$\mathcal{D}^* = \underset{\mathcal{D} \in \mathbb{D}}{\arg\min} \mathcal{M}(\hat{f}, f^*) + \lambda \cdot \text{card}(\mathcal{D})$$

$$\text{s.t.} \quad \hat{f} = \mathcal{A}(\mathcal{D}). \quad (3)$$

The formulation above involves three key components: $\mathcal{M}$ which quantifies the disagreement between $\hat{f}$ and $f^*$ (*e.g.*, $L_2$ distance in RKHS $\mathcal{M}(\hat{f}^*, f^*) = \|\hat{f}^* - f^*\|_{\mathcal{H}}$), $\text{card}(\cdot)$, which represents the cardinality of the teaching sequence $\mathcal{D}$, regularized by a constant $\lambda$, and $\mathcal{A}(\mathcal{D})$, which refers to the learning algorithm used by the learners, typically employing empirical risk minimization:

$$\hat{f} = \underset{f \in \mathcal{H}}{\arg\min} \mathbb{E}_{\boldsymbol{x} \sim \mathbb{P}(\boldsymbol{x})} \left(\mathcal{L}(f(\boldsymbol{x}), f^*(\boldsymbol{x}))\right) \quad (4)$$

with a convex (w.r.t. $f$) loss $\mathcal{L}$, which is optimized using functional gradient descent:

$$f^{t+1} \leftarrow f^t - \eta \mathcal{G}(\mathcal{L}, f^*; f^t, \boldsymbol{x}^t), \quad (5)$$

where $t = 0, 1, \dots, T$ is the time step, $\eta > 0$ represents the learning rate, and $\mathcal{G}$ denotes the functional gradient computed at time $t$.

To derive the functional gradient, given by

$$\mathcal{G}(\mathcal{L}, f^*; f^\dagger, \boldsymbol{x}) = E_{\boldsymbol{x}} \left(\left.\frac{\partial \mathcal{L}(f^*, f)}{\partial f}\right|_{f^\dagger}\right) \cdot K_{\boldsymbol{x}}, \quad (6)$$

Zhang et al., 2023b;a introduce the chain rule for functional gradients (Gelfand et al., 2000) (see Lemma 3) and use the Fréchet derivative to compute the derivative of the evaluation functional in RKHS (Coleman, 2012) (cf. Lemma 4).

**Lemma 3.** *(Chain rule for functional gradients) For differentiable functions $G(F) : \mathbb{R} \mapsto \mathbb{R}$ that depend on functionals $F(f) : \mathcal{H} \mapsto \mathbb{R}$, the expression*

$$\nabla_f G(F(f)) = \frac{\partial G(F(f))}{\partial F(f)} \cdot \nabla_f F(f) \quad (7)$$

*is typically referred to as the chain rule.*

**Lemma 4.** *The gradient of the evaluation functional at the feature $\boldsymbol{x}$, defined as $E_{\boldsymbol{x}}(f) = f(\boldsymbol{x}) : \mathcal{H} \to \mathbb{R}$, is given by $\nabla_f E_{\boldsymbol{x}}(f) = K(\boldsymbol{x}, \cdot)$, where $K(\boldsymbol{x}, \boldsymbol{x}') : \mathbb{R}^d \times \mathbb{R}^d \to \mathbb{R}$ represents a feature-based kernel.*

## 4. GraNT

We begin by analyzing the effect of the adjacency matrix on parameter-based gradient descent. Then, by translating the evolution of GCN—driven by structure-aware updates in parameter space—into function space, we show that the evolution of GCN under parameter gradient descent aligns with that under functional gradient descent. Lastly, we present the greedy GraNT algorithm, which efficiently selects graphs with steeper gradients to improve the learning efficiency of GCN.

### 4.1. Structure-aware update in the parameter space

In GCNs, the structural information of graphs is captured through feature aggregation, expressed as $(\boldsymbol{A} + \boldsymbol{I})^\kappa \boldsymbol{X}$, as shown in Equation 2. The use of $(\boldsymbol{A} + \boldsymbol{I})^\kappa$ limits the flexibility in learning aggregated hidden features, $\sigma((\boldsymbol{A} + \boldsymbol{I})^\kappa \boldsymbol{X} \boldsymbol{W})$, because it applies the same weights to features aggregated from different convolutional orders within a single layer. This paper considers more flexible GCNs, where the weights for features aggregated from different convolutional orders within a single layer are handled independently (Krishnagopal & Ruiz, 2023). Before presenting the detailed formulation, we introduce the concatenation operation $\bigoplus$ and define $\boldsymbol{A}^{[\kappa]} := \bigoplus_{i=0}^{\kappa-1} \boldsymbol{A}^i = [\boldsymbol{I} \, \boldsymbol{A} \, \cdots \, \boldsymbol{A}^{\kappa-1}]$, an $n \times \kappa n$ matrix. By unfolding the aggregated features at different orders and assigning them distinct weights (Krishnagopal & Ruiz, 2023), the flexible GCN can be expressed as

$$\boldsymbol{X}^{(\ell)} = \sigma\left(\boldsymbol{A}^{[\kappa_\ell]} \text{diag}(\boldsymbol{X}^{(\ell-1)}; \kappa_\ell) \cdot \boldsymbol{W}^{(\ell)}\right), \ell \in \mathbb{N}_{L-1}$$

$$\boldsymbol{X}^{(L)} = \rho\left(\boldsymbol{A}^{[\kappa_L]} \text{diag}(\boldsymbol{X}^{(L-1)}; \kappa_L) \cdot \boldsymbol{W}^{(L)}\right). \quad (8)$$

Here, the notations are consistent with those in Equation 2, with the exception that $\boldsymbol{W}^{(\ell)}$ is the weight matrix of size $\kappa_\ell h_{\ell-1} \times h_\ell$. Figure 2 presents an example that illustrates the workflow of this flexible GCN.

Let the column vector $\theta \in \mathbb{R}^m$ represent the weights of all layers in a flattened form, where $m$ denotes the total number of parameters in the GCN. Given a training set of size $N$,

$\{(\boldsymbol{G}_i, \boldsymbol{y}_i)|\boldsymbol{G}_i \in \mathbb{G}, \boldsymbol{y}_i \in \mathcal{Y}\}_N$, the parameter update using gradient descent (Ruder, 2016) is expressed as follows:

$$\theta^{t+1} \leftarrow \theta^t - \frac{\eta}{N}\sum_{i=1}^{N}\nabla_\theta \mathcal{L}(f_{\theta^t}(\boldsymbol{G}_i), \boldsymbol{y}_i). \quad (9)$$

Due to the sufficiently small learning rate $\eta$, the updates are tiny over several iterations, which allows them to be treated as a time derivative and then converted into a differential equation (Du et al., 2019):

$$\frac{\partial \theta^t}{\partial t} = -\frac{\eta}{N}\left[\frac{\partial \mathcal{L}(f_{\theta^t}(\boldsymbol{G}_i), \boldsymbol{y}_i)}{\partial f_{\theta^t}(\boldsymbol{G}_i)}\right]_N^\top \cdot \left[\frac{\partial f_{\theta^t}(\boldsymbol{G}_i)}{\partial \theta^t}\right]_N. \quad (10)$$

The term $\frac{\partial f_\theta(\boldsymbol{G})}{\partial \theta}$ (with the indexes omitted for simplicity), which indicates the direction for updating the parameters, can be written more specifically as

$$\frac{\partial f_\theta(\boldsymbol{G})}{\partial \theta} = \left[\frac{\partial \boldsymbol{X}^{(L)}}{\partial \boldsymbol{W}^{(L)}}, \underbrace{\frac{\partial \boldsymbol{X}^{(L)}}{\partial \boldsymbol{W}^{(L-1)}_{(:,1)}}, \cdots, \frac{\partial \boldsymbol{X}^{(L)}}{\partial \boldsymbol{W}^{(L-1)}_{(:,h_{L-1})}}}_{\text{w.r.t. the } (L-1)\text{-th layer}},\right.$$

$$\left.\cdots, \underbrace{\frac{\partial \boldsymbol{X}^{(L)}}{\partial \boldsymbol{W}^{(1)}_{(:,1)}}, \cdots, \frac{\partial \boldsymbol{X}^{(L)}}{\partial \boldsymbol{W}^{(1)}_{(:,h_1)}}}_{\text{w.r.t. the first layer}}\right]. \quad (11)$$

Here, each term represents the derivative of output $\boldsymbol{X}^{(L)}$ w.r.t. weight column vectors. In contrast to derivatives with regular features as inputs, where the derivatives are independent across features, the adjacency matrix $\boldsymbol{A}$ dictates feature aggregation in these derivatives in a batch manner, where each feature of a single node is treated individually (Du et al., 2019). To clearly demonstrate, in an analytical and explicit manner, how $\boldsymbol{A}$ directs structure-aware updates in the parameter space, we present an example involving the derivative of a two-layer GCN with summation pooling:

$$\frac{\partial f_\theta(\boldsymbol{G})}{\partial \theta} = \left[\frac{\partial \boldsymbol{X}^{(2)}}{\partial \boldsymbol{W}^{(2)}}, \frac{\partial \boldsymbol{X}^{(2)}}{\partial \boldsymbol{W}^{(1)}_{(:,1)}}, \cdots, \frac{\partial \boldsymbol{X}^{(2)}}{\partial \boldsymbol{W}^{(1)}_{(:,h_1)}}\right], \quad (12)$$

where the term $\frac{\partial \boldsymbol{X}^{(2)}}{\partial \boldsymbol{W}^{(2)}}$ is given by

$$\underbrace{\underbrace{\mathbf{1}_n^\top \underbrace{\boldsymbol{A}^{[\kappa_2]}}_{\text{size: } n \times \kappa_2 n}}_{\text{size: } 1 \times \kappa_2 h_1} \overbrace{\mathrm{diag}(\sigma(\boldsymbol{A}^{[\kappa_1]}\mathrm{diag}(\boldsymbol{X}^{(0)};\kappa_1)\boldsymbol{W}^{(1)});\kappa_2)}^{\text{size: } \kappa_2 n \times \kappa_2 h_1}}, \quad (13)$$

and for $i \in \mathbb{N}_{h_1}$, the term $\frac{\partial \boldsymbol{X}^{(2)}}{\partial \boldsymbol{W}^{(1)}_{(:,i)}}$ is

$$\underbrace{\mathbf{1}^\top \underbrace{\boldsymbol{A}^{[\kappa_2]}}_{\text{size: } n \times \kappa_2 n} \overbrace{\begin{pmatrix} \dot{\sigma} \cdot \boldsymbol{A}^{[\kappa_1]}\mathrm{diag}(\boldsymbol{X}^{(0)};\kappa_1) \cdot \boldsymbol{W}^{(2)}_{(i-h_1+h_1)} \\ \cdots \\ \dot{\sigma} \cdot \boldsymbol{A}^{[\kappa_1]}\mathrm{diag}(\boldsymbol{X}^{(0)};\kappa_1) \cdot \boldsymbol{W}^{(2)}_{(i-h_1+\kappa_2 h_1)} \end{pmatrix}}^{\text{size: } \kappa_2 n \times \kappa_1 h_0}}_{\text{size: } 1 \times \kappa_1 h_0}, \quad (14)$$

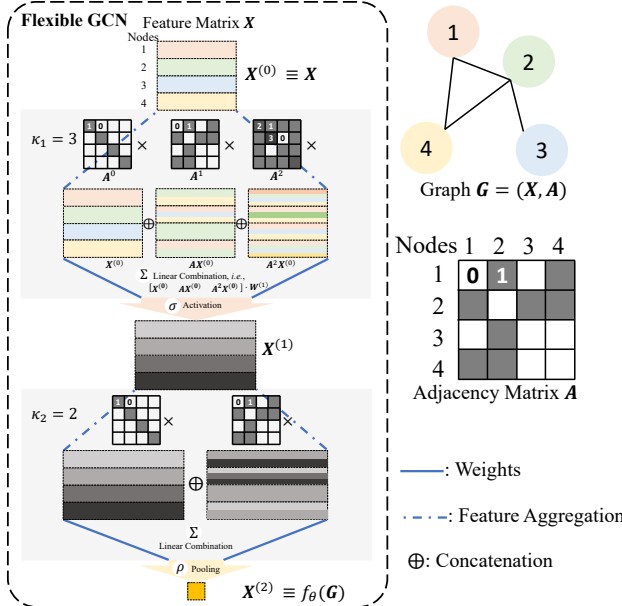

Figure 2: A workflow illustration of a two-layer flexible GCN with a four-node graph $\boldsymbol{G}$ as input.

with $\dot{\sigma} = \frac{\partial \sigma(\boldsymbol{A}^{[\kappa_1]}\mathrm{diag}(\boldsymbol{X}^{(0)};\kappa_1)\boldsymbol{W}^{(1)}_{(:,i)})}{\partial \boldsymbol{A}^{[\kappa_1]}\mathrm{diag}(\boldsymbol{X}^{(0)};\kappa_1)\boldsymbol{W}^{(1)}_{(:,i)}}$. Orange indicates the first layer, green denotes the second layer, and $\mathbf{1}_n^\top$ in purple corresponds to the summation pooling[4]. When using ReLU as the activation function, $\dot{\sigma} \cdot \boldsymbol{A}^{[\kappa_1]}\mathrm{diag}(\boldsymbol{X}^{(0)};\kappa_1) = \sigma\left(\boldsymbol{A}^{[\kappa_1]}\mathrm{diag}(\boldsymbol{X}^{(0)};\kappa_1)\right)$. The derivation can be found in Appendix A.2.

When the convolutional order $\kappa$ is reduced to 1 for all layers, meaning structural information is excluded, the GCN gradient computed for a single input graph exactly matches that of the MLP when applied to a batch composed of the node features. This suggests that the structure-aware, parameter-based gradient is more general than the one used in MLPs, indicating that this work can be seen as a generalization of Zhang et al., 2024a. Furthermore, from the explicit expressions in Equations 13 and 14, it can be observed that the gradient of GCN does not depend on the size of the input graph (i.e., the number of nodes). Instead, it depends on the feature dimension and convolutional order. In other words, the parameter gradient retains the same form even when the input graph size $n$ is scaled.

## 4.2. The functional evolution of GCN

The structure-aware update in the parameter space drives the functional evolution of $f_\theta \in \mathcal{H}$. This variation of $f_\theta$, which captures how $f_\theta$ changes in response to updates in $\theta$, can be derived using Taylor's theorem as follows:

$$f(\theta^{t+1}) - f(\theta^t) = \langle \nabla_\theta f(\theta^t), \theta^{t+1} - \theta^t \rangle + o(\theta^{t+1} - \theta^t), \quad (15)$$

[4]Without the pooling operation, it corresponds to the scenario where the graph property is considered at the node level.

where $f(\theta^\dagger) \equiv f_{\theta^\dagger}$. In a manner similar to the transformation of parameter updates, it can be rewritten in a differential form (Zhang et al., 2024a):

$$\frac{\partial f_{\theta^t}}{\partial t} = \underbrace{\left\langle \frac{\partial f(\theta^t)}{\partial \theta^t}, \frac{\partial \theta^t}{\partial t} \right\rangle}_{(*)} + o\left(\frac{\partial \theta^t}{\partial t}\right). \quad (16)$$

By plugging in the specific parameter updates, *i.e.*, Equation 10, into the first-order approximation term $(*)$ of this variation, we get

$$\frac{\partial f_{\theta^t}}{\partial t} = -\frac{\eta}{N} \left[\frac{\partial \mathcal{L}(f_{\theta^t}(\boldsymbol{G}_i), \boldsymbol{y}_i)}{\partial f_{\theta^t}(\boldsymbol{G}_i)}\right]_N^\top \cdot [K_{\theta^t}(\boldsymbol{G}_i, \cdot)]_N + o\left(\frac{\partial \theta^t}{\partial t}\right), \quad (17)$$

where the symmetric and positive definite $K_{\theta^t}(\boldsymbol{G}_i, \cdot) \coloneqq \left\langle \frac{\partial f_{\theta^t}(\boldsymbol{G}_i)}{\partial \theta^t}, \frac{\partial f_{\theta^t}(\cdot)}{\partial \theta^t} \right\rangle$ (see the detailed derivation in Appendix A.3). Due to the inclusion of nonlinear activation functions in $f(\theta)$, the nonlinearity of $f(\theta)$ with respect to $\theta$ causes the remainder $o(\theta^{t+1} - \theta^t)$ to be nonzero. In a subtle difference, Jacot et al., 2018; Du et al., 2019; Krishnagopal & Ruiz, 2023 apply the chain rule directly, giving less attention to the convexity of $\mathcal{L}$ with respect to $\theta$. This leads to the first-order approximation being derived as the variation, with $K_\theta$ being referred to as the graph neural tangent kernel (GNTK). It has been shown that the GNTK stays constant during training when the GCN width is assumed to be infinite (Du et al., 2019; Krishnagopal & Ruiz, 2023). However, in practical applications, there is no need for the GCN width to be infinitely large, which leads us to investigate the dynamic GNTK (Figure 5 in Appendix A.3 provides an example of how GNTK is computed).

Consider describing the variation of $f_\theta \in \mathcal{H}$ from a high-level, functional perspective (Zhang et al., 2024a). Using functional gradient descent, it can be expressed as:

$$\frac{\partial f_{\theta^t}}{\partial t} = -\eta \mathcal{G}(\mathcal{L}, f^*; f_{\theta^t}, \{\boldsymbol{G}_i\}_N), \quad (18)$$

where the functional gradient is given by:

$$\mathcal{G}(\mathcal{L}, f^*; f_{\theta^t}, \{\boldsymbol{G}_i\}_N) = \frac{1}{N}\left[\frac{\partial \mathcal{L}(f_{\theta^t}(\boldsymbol{G}_i), \boldsymbol{y}_i)}{\partial f_{\theta^t}(\boldsymbol{G}_i)}\right]_N^\top \cdot [K(\boldsymbol{G}_i, \cdot)]_N. \quad (19)$$

The asymptotic relationship between GNTK and the structure-aware canonical kernel (Vishwanathan et al., 2010; Zhang et al., 2024a) in the context of functional gradient is given in Theorem 5 below, with the proof in Appendix B.

**Theorem 5.** *For a convex loss $\mathcal{L}$ and a given training set $\{(\boldsymbol{G}_i, \boldsymbol{y}_i) | \boldsymbol{G}_i \in \mathbb{G}, \boldsymbol{y}_i \in \mathcal{Y}\}_N$, the dynamic GNTK, derived from gradient descent on the parameters of a GCN, converges pointwise to the structure-aware canonical kernel in the dual functional gradient with respect to the input graphs. Specifically, the following holds:*

$$\lim_{t\to\infty} K_{\theta^t}(\boldsymbol{G}_i, \cdot) = K(\boldsymbol{G}_i, \cdot), \forall i \in \mathbb{N}_N. \quad (20)$$

This suggests that GNTK, which incorporates structural information, serves as a dynamic alternative to the structure-aware canonical kernel in functional gradient descent with graph inputs, making the GCN's evolution through parameter gradient descent align with that in functional gradient descent (Kuk, 1995; Du et al., 2019; Geifman et al., 2020). This functional insight bridges the teaching of the graph property learner, GCN, with that of structure-aware nonparametric learners, while also simplifying further analysis (*e.g.*, a convex functional $\mathcal{L}$ preserves its convexity with respect to $f_\theta$ from a functional perspective, but is typically nonconvex when considering $\theta$). By leveraging the functional insight and employing the canonical kernel (Dou & Liang, 2021) instead of GNTK (which should be considered *alongside the remainder*), it aids in deriving sufficient reduction concerning $\mathcal{L}$ in Proposition 6, with the proof deferred to Appendix B.

**Proposition 6.** *(Sufficient Loss Reduction) Suppose the convex loss $\mathcal{L}$ is Lipschitz smooth with a constant $\tau > 0$, and the structure-aware canonical kernel is bounded above by a constant $\gamma > 0$. If the learning rate $\eta$ satisfies $\eta \leq 1/(2\tau\gamma)$, then it follows that a sufficient reduction in $\mathcal{L}$ is guaranteed, as shown by*

$$\frac{\partial \mathcal{L}}{\partial t} \leq -\frac{\eta\gamma}{2}\left(\frac{1}{N}\sum_{i=1}^N \frac{\partial \mathcal{L}(f_{\theta^t}(\boldsymbol{G}_i), \boldsymbol{y}_i)}{\partial f_{\theta^t}(\boldsymbol{G}_i)}\right)^2. \quad (21)$$

This demonstrates that the variation of $\mathcal{L}$ over time is capped by a negative value, meaning it decreases by at least the magnitude of this upper bound as time progresses, guaranteeing convergence.

### 4.3. GraNT algorithm

Building on the insights regarding the effect of the adjacency matrix, which captures the graph structure, on parameter-based gradient descent, as well as the consistency between teaching a GCN and a nonparametric learner, we introduce the GraNT algorithm. This algorithm seeks to increase the steepness of gradients to improve the learning efficiency of GCN. By interpreting the gradient as a sum of the projections of $\frac{\partial \mathcal{L}(f_\theta, f^*)}{\partial f_\theta}$ onto the basis $\{K(\boldsymbol{G}_i, \cdot)\}_N$, increasing the gradient can be achieved by simply maximizing the projection coefficient $\frac{\partial \mathcal{L}(f_\theta(\boldsymbol{G}_i), \boldsymbol{y}_i)}{\partial f_\theta(\boldsymbol{G}_i)}$, without the need to calculate the norm of the basis $\|K(\boldsymbol{G}_i, \cdot)\|_{\mathcal{H}}$ (Wright, 2015; Zhang et al., 2024a). This suggests that selecting graphs that either maximize $\left|\frac{\partial \mathcal{L}(f_\theta(\boldsymbol{G}_i), \boldsymbol{y}_i)}{\partial f_\theta(\boldsymbol{G}_i)}\right|$ or correspond to larger components of $\frac{\partial \mathcal{L}(f_\theta, f^*)}{\partial f_\theta}$ can effectively increase the gradient, which implies

$$\{\boldsymbol{G}_i\}_m^* = \operatorname*{arg\,max}_{\{\boldsymbol{G}_i\}_m \subseteq \{\boldsymbol{G}_i\}_N} \left\|\left[\frac{\partial \mathcal{L}(f_\theta(\boldsymbol{G}_i), \boldsymbol{y}_i)}{\partial f_\theta(\boldsymbol{G}_i)}\right]_m\right\|_2. \quad (22)$$

---

**Algorithm 1** GraNT Algorithm

---

**Input:** Target mapping $f^*$ realized by a dense set of graph-property pairs, initial GCN $f_{\theta^0}$, the size of selected training set $m \leq N$, small constant $\epsilon > 0$ and maximal iteration number $T$.

Set $f_{\theta^t} \leftarrow f_{\theta^0}, t = 0$.

**while** $t \leq T$ and $\|[f_{\theta^t}(\boldsymbol{G}_i) - f^*(\boldsymbol{G}_i)]_N\|_2 \geq \epsilon$ **do**

    **The teacher** selects $m$ teaching graphs:

    `/* (`**`Graph-level`**`) Graphs corresponding`
    `   to the m largest` $|f_{\theta^t}(\boldsymbol{G}_i) - f^*(\boldsymbol{G}_i)|.$ `*/`
    $\{\boldsymbol{G}_i\}_m{}^* = \underset{\{\boldsymbol{G}_i\}_m \subseteq \{\boldsymbol{G}_i\}_N}{\arg\max} \|[f_{\theta^t}(\boldsymbol{G}_i) - f^*(\boldsymbol{G}_i)]_m\|_2.$

    `/* (`**`Node-level`**`) Graphs associated with`
    `   the m largest` $\frac{\|f_{\theta^t}(\boldsymbol{G}_i) - f^*(\boldsymbol{G}_i)\|_2}{n_i}.$ `*/`
    $\{\boldsymbol{G}_i\}_m{}^* = \underset{\{\boldsymbol{G}_i\}_m \subseteq \{\boldsymbol{G}_i\}_N}{\arg\max} \left\|\left[\frac{f_{\theta^t}(\boldsymbol{G}_i) - f^*(\boldsymbol{G}_i)}{n_i}\right]_m\right\|_{\mathcal{F}},$
    with Frobenius norm $\|\cdot\|_{\mathcal{F}}$.

    Provide $\{\boldsymbol{G}_i\}_m{}^*$ to the GCN learner.

    **The learner** updates $f_{\theta^t}$ based on received $\{\boldsymbol{G}_i\}_m{}^*$:

    `// Parameter-based gradient descent.`
    $\theta^t \leftarrow \theta^t - \frac{\eta}{m} \sum_{\boldsymbol{G}_i \in \{\boldsymbol{G}_i\}_m{}^*} \nabla_\theta \mathcal{L}(f_{\theta^t}(\boldsymbol{G}_i), f^*(\boldsymbol{G}_i)).$

    Set $t \leftarrow t + 1$.

**end**

---

From a functional standpoint, when handling a convex loss functional $\mathcal{L}$, the norm of the partial derivative of $\mathcal{L}$ with respect to $f_\theta$, represented as $\|\frac{\partial \mathcal{L}(f_\theta)}{\partial f_\theta}\|_{\mathcal{H}}$, is positively correlated with $\|f_\theta - f^*\|_{\mathcal{H}}$. As $f_\theta$ progressively converges to $f^*$, $\|\frac{\partial \mathcal{L}(f_\theta)}{\partial f_\theta}\|_{\mathcal{H}}$ diminishes (Boyd et al., 2004; Coleman, 2012). This relationship becomes especially noteworthy when $\mathcal{L}$ is strongly convex with a larger convexity constant (Kakade & Tewari, 2008; Arjevani et al., 2016). Leveraging these insights, the GraNT algorithm selects graphs by

$$\{\boldsymbol{G}_i\}_m{}^* = \underset{\{\boldsymbol{G}_i\}_m \subseteq \{\boldsymbol{G}_i\}_N}{\arg\max} \|[f_\theta(\boldsymbol{G}_i) - f^*(\boldsymbol{G}_i)]_m\|_2. \quad (23)$$

The pseudo code, including the node-level version, is provided in Algorithm 1.

## 5. Experiments and Results

We start by evaluating GraNT on graph-level regression and classification tasks, then proceed to validate it on node-level tasks. The overall results on the test set are shown in Table 1, which clearly highlights the effectiveness of GraNT in graph property learning: it reduces training time by 36.62% for graph-level regression, 38.19% for graph-level classification, 30.97% for node-level regression, and 47.30% for node-level classification, all while maintaining

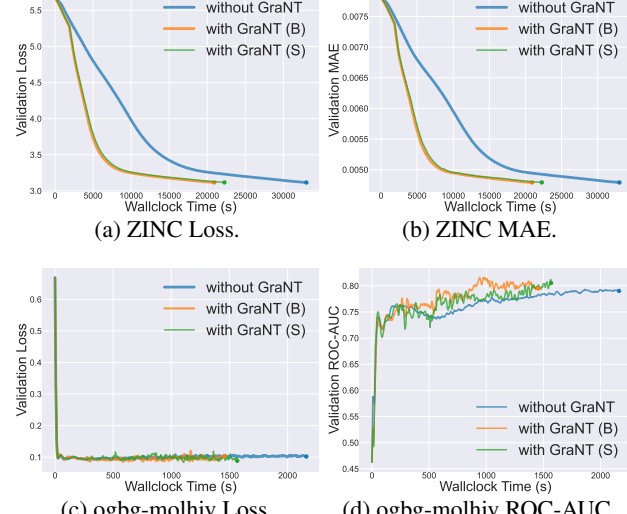

(a) ZINC Loss.  (b) ZINC MAE.

(c) ogbg-molhiv Loss.  (d) ogbg-molhiv ROC-AUC.

Figure 3: Validation set performance for graph-level tasks: ZINC (regression) and ogbg-molhiv (classification).

comparable testing performance. Detailed settings are given in Appendix C.

Given the common practice of training GCN in batches, *i.e.*, graphs are fed in batches, it is both natural and intuitive to implement GraNT at the batch level. This involves selecting batches that exhibit the largest average discrepancy between the actual properties and the corresponding GCN outputs, referred to as GraNT (B). Meanwhile, another variant, called GraNT (S), selects single graph with the largest discrepancies within each batch in proportion, then reorganizes the selected graphs into new batches.

**Graph-level tasks.** We evaluate GraNT using several widely recognized benchmark datasets as follows:

- QM9 (Wu et al., 2018): 130k organic molecules graphs with quantum chemical properties (regression task);

- ZINC (Gómez-Bombarelli et al., 2018): 250k molecular graphs with bioactivity and solubility chemical properties (regression task);

- ogbg-molhiv (Hu et al., 2020): 41k molecular graphs with HIV inhibitory activity properties (binary classification task);

- ogbg-molpcba (Hu et al., 2020): 438k molecular graphs with bioactivity properties (multi-task binary classification task).

To clearly illustrate the practical efficiency of GraNT, we plot the wallclock time versus loss/metric curves. This is done by conducting a validation after each training epoch, *i.e.*, performing an evaluation on the validation dataset after each training process. Specifically, we display the validation set loss and the typical Mean Absolute Error (MAE) for ZINC in Figure 3 (a) and (b), respectively. In both plots,

| GraNT | | Dataset | Time (s) | Loss ↓ | MAE ↓ | ROC-AUC ↑ | AP ↑ |
|---|---|---|---|---|---|---|---|
| ✗ | | QM9 | 9654.81 | 2.0444 | 0.0051±0.0009 | - | - |
| | | ZINC | 33033.82 | 3.1160 | 0.0048±0.0004 | - | - |
| | | ogbg-molhiv | 2163.50 | 0.1266 | - | 0.7572±0.0005 | - |
| | | ogbg-molpcba | 130191.26 | 0.0577 | - | - | 0.3270±0.0000 |
| | | gen-reg | 3344.78 | 0.0086 | 0.0007±0.0001 | - | - |
| | | gen-cls | 11662.25 | 0.1314 | - | 0.9150±0.0024 | - |
| ✓ | B | QM9 | 6392.26 (-33.79%) | 2.0436 | 0.0051±0.0009 | - | - |
| | | ZINC | 20935.24 (-36.62%) | 3.1165 | 0.0048±0.0004 | - | - |
| | | ogbg-molhiv | 1457.39 (-32.64%) | 0.1238 | - | 0.7676±0.0036 | - |
| | | ogbg-molpcba | 80465.06 (-38.19%) | 0.0577 | - | - | **0.3358±0.0001** |
| | | gen-reg | 2308.97 (-30.97%) | 0.0086 | 0.0007±0.0001 | - | - |
| | | gen-cls | 6145.72 (-47.30%) | 0.1314 | - | **0.9157±0.0013** | - |
| | S | QM9 | 7076.37 (-26.71%) | 2.0443 | 0.0051±0.0009 | - | - |
| | | ZINC | 22265.83 (-32.60%) | 3.1170 | 0.0048±0.0004 | - | - |
| | | ogbg-molhiv | 1597.69 (-26.15%) | 0.1421 | - | **0.7705±0.0027** | - |
| | | ogbg-molpcba | 89858.65 (-30.98%) | 0.0575 | - | - | 0.3351±0.0025 |
| | | gen-reg | 2337.46 (-30.12%) | 0.0086 | 0.0007±0.0001 | - | - |
| | | gen-cls | 8171.21 (-29.93%) | 0.1313 | - | **0.9157±0.0014** | - |

Table 1: Training time and testing results across different benchmarks. GraNT (B) and GraNT (S) demonstrate similar testing performance while significantly reducing training time compared to the "without GraNT", across graph-level (QM9, ZINC, ogbg-molhiv, ogbg-molpcba) and node-level (gen-reg, gen-cls) datasets, for both regression and classification tasks.

| | Time (s) | MAE ↓ |
|---|---|---|
| AL-3DGraph[‡] (Subedi et al., 2024) | 9200.27 | 0.7991 |
| AL-3DGraph[♯] (Subedi et al., 2024) | 9364.74 | 0.4719 |
| AL-3DGraph[§] (Subedi et al., 2024) | 12601.77 | 0.1682 |
| GraNT (B) | **6392.26** | **0.0051** |
| GraNT (S) | 7076.37 | **0.0051** |

[‡]: lr=5e-5, batch_size=256, which matches GraNT settings.
[♯]: lr=5e-4, batch_size=256.
[§]: lr=5e-4, batch_size=32, which corresponds to the default settings used in the provided code for that paper.

Table 2: Comparison of GraNT with active learning-based methods on the QM9 dataset.

| | Time (s) | ROC-AUC ↑ |
|---|---|---|
| GCN (Kipf & Welling, 2017) | 2888.80 | 0.7385 |
| GCN+Virtual Node (Kipf & Welling, 2017) | 3083.16 | 0.7608 |
| GMoE-GCN (Wang et al., 2023) | 3970.16 | 0.7536 |
| GMoE-GIN (Wang et al., 2023) | 3932.06 | 0.7468 |
| GDeR-GCN[†] (Zhang et al., 2024b) | 1772.23 | 0.7261 |
| GDeR-PNA[†] (Zhang et al., 2024b) | 5088.88 | 0.7616 |
| GraNT (B) | **1457.39** | 0.7676 |
| GraNT (S) | 1597.69 | **0.7705** |

[†]: batch_size=500, retain_ratio=0.7.

Table 3: Comparison of GraNT with recent efficient methods on the ogbg-molhiv dataset.

one can see that the curves for GraNT (B) and GraNT (S) span about two-thirds of the width of the "without GraNT" curve, with both the loss and MAE for GraNT decreasing at a faster rate than those for the "without GraNT" case. Moreover, GraNT (B) takes slightly less time to terminate than GraNT (S). This is because GraNT (S) selects teaching graphs from each batch, which can add extra operational time compared to GraNT (B) that uses direct batch selection.

Figures 3 (c) and (d) show the loss and the commonly used ROC-AUC curves on the validation set, respectively, for ogbg-molhiv. Both plots clearly highlight the superiority of GraNT over the "without GraNT". In addition, Figure 3 (d) shows that the ROC-AUC values of GraNT (B) and GraNT (S) consistently exceed that of "without GraNT"

once the wallclock time reaches approximately 500s. However, the curves appear relatively jagged, which can be attributed to the label imbalance in this benchmark dataset. This imbalance also explains why, even when the validation loss decreases significantly, the ROC-AUC curve does not rise to a higher range. The detailed numerical results for training time and testing performance are provided in Table 1. The comparisons between GraNT and recent SOTA methods are shown in Table 2 for QM9 and Table 3 for ogbg-molhiv.

**Node-level tasks.** We also assess GraNT for node-level property learning using synthetic data. Specifically, we utilize the graphon, a typical limit object of a convergent sequence of graphs (Xu et al., 2021; Xia et al., 2023), to generate two synthetic datasets: gen-reg (containing 50k

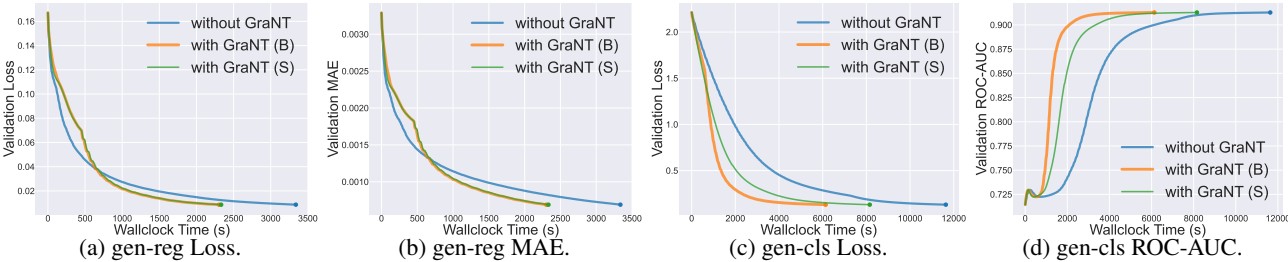

(a) gen-reg Loss.     (b) gen-reg MAE.     (c) gen-cls Loss.     (d) gen-cls ROC-AUC.

Figure 4: Validation set performance for node-level tasks: gen-reg (regression) and gen-cls (classification).

graphs) for regression and gen-cls (containing 50k graphs) for classification.

Figures 4 (a) and (b) illustrate the validation loss and MAE curves for gen-reg, respectively. From both plots, it is clear that GraNT reached convergence more quickly in terms of wallclock time compared to the "without GraNT", highlighting its efficiency.

Figure 4 (c) and (d) show the validation loss and ROC-AUC for gen-cls, respectively. Both plots demonstrate that GraNT requires less wallclock time to converge compared to the "without GraNT". Furthermore, although this dataset is generated with imbalanced labels, similar to ogbg-molhiv, there is a notable difference: when the validation loss is low, the corresponding ROC-AUC exceeds 0.9, which is a relatively high value. This underscores the effectiveness of GraNT. The detailed numerical results for training time and testing performance are shown in Table 1.

All experimental results demonstrate that GraNT (B) and GraNT (S) offer significant time-saving benefits while maintaining comparable generalization performance, and in some cases, even outperforming the "without GraNT". Further experimental results, including result plots for QM9 and ogbg-molpcba, training curves for the aforementioned datasets, and additional validations on the AMD device, can be found in Appendix C.

## 6. Concluding Remarks and Future Work

This paper proposes GraNT, a novel paradigm that enhances the learning efficiency of graph property learner (GCN) through nonparametric teaching theory. Specifically, GraNT reduces the wallclock time needed to learn the implicit mapping from graphs to properties of interest by over 30%, while maintaining comparable test performance, as shown through extensive experiments. Furthermore, GraNT establishes a theoretical connection between the evolution of a GCN using parameter-based gradient descent and that of a function using functional gradient descent in nonparametric teaching. This connection between nonparametric teaching theory and GCN training broadens the potential applications of nonparametric teaching in graph property learning.

In future work, it would be interesting to investigate other

variations of GraNT for different graph property learners, such as graph attention networks (Veličković et al., 2018). Moreover, exploring the practical applications of GraNT to improve the efficiency of data-driven methods (Henaff, 2020; Touvron et al., 2021; Müller et al., 2022) within the field of graph property learning offers promising opportunities for future progress, particularly in fields like molecular biology and protein research.

## Impact Statement

Recent interest in learning implicit mappings from graph data to specific properties has grown significantly, especially in science-related fields, driven by the ability of graphs to model diverse types of data. This work focuses on improving the learning efficiency of implicit graph property mappings through a novel nonparametric teaching perspective, which has the potential to positively impact graph-related fields and society. Furthermore, it connects nonparametric teaching theory with GCN training, expanding its application in graph property learning. As a result, it also holds promise for valuable contributions to the nonparametric teaching community.

## Acknowledgements

We thank all anonymous reviewers for their constructive feedback, which helped improve our paper. We also thank Yikun Wang for his helpful discussions. This work is supported in part by the Theme-based Research Scheme (TRS) project T45-701/22-R of the Research Grants Council (RGC), Hong Kong SAR, and in part by the AVNET-HKU Emerging Microelectronics & Ubiquitous Systems (EMUS) Lab.

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

# Appendix

## A. Additional Discussions

### A.1. Notation Overview

| Notation | Description |
|---|---|
| $\boldsymbol{G}_{(n)} = (\mathcal{V}, \mathcal{E})$ | Graph with $n$ vertices and edge set $\mathcal{E}$ |
| $\mathcal{V}$ | Set of $n$ vertices (nodes) in the graph |
| $\mathcal{E}$ | Set of edges in the graph |
| $[x_i]_d$ | $d$-dimensional feature vector with entries $x_i$ |
| $\boldsymbol{x}$ | Simplified notation for $[x_i]_d$ |
| $\boldsymbol{X}_{n \times d}$ | Feature matrix of all nodes ($n \times d$) |
| $\boldsymbol{X}_{(i,:)}$ | $i$-th row of $\boldsymbol{X}$ (feature vector of node $i$) |
| $\boldsymbol{X}_{(:,j)}$ | $j$-th column of $\boldsymbol{X}$ (feature $j$ across nodes) |
| $\boldsymbol{e}_i$ | $i$-th basis vector (1 at $i$-th position, 0 elsewhere) |
| $\boldsymbol{A}$ | Adjacency matrix of graph $\boldsymbol{G}_{(n)}$ |
| $\boldsymbol{G} = (\boldsymbol{X}, \boldsymbol{A})$ | Representation of graph with feature matrix and adjacency matrix |
| $\mathbb{G}$ | Collection of all graphs |
| $\boldsymbol{y}$ | Property of the graph (scalar or vector) |
| $\mathcal{Y}$ | Space of graph properties ($\mathbb{R}$ or $\mathbb{R}^n$) |
| $\{a_i\}_m$ | Set of $m$ elements |
| $\mathrm{diag}(a_1, \ldots, a_m)$ | Diagonal matrix with elements $a_1, \ldots, a_m$ |
| $\mathrm{diag}(a; m)$ | Diagonal matrix with $m$ repeated entries $a$ |
| $\mathbb{N}_d := \{1, \cdots, d\}$ | Set of natural numbers from 1 to $d$ |
| $K(\boldsymbol{G}, \boldsymbol{G}')$ | Positive definite graph kernel function |
| $\mathcal{H}$ | Reproducing kernel Hilbert space (RKHS) defined by $K$ |
| $f^*$ | Target mapping from $\mathbb{G}$ to $\mathcal{Y}$ |
| $\boldsymbol{y}_\dagger$ | Property $f^*(\boldsymbol{G}_\dagger)$ of graph $\boldsymbol{G}_\dagger$ |

Table 4: Summary of Key Notations.

### A.2. The derivation of structure-aware updates in the parameter space.

Let's focus on the derivative of a two-layer GCN with summation pooling:

$$\frac{\partial f_\theta(\boldsymbol{G})}{\partial \theta} = \left[ \underbrace{\frac{\partial \boldsymbol{X}^{(2)}}{\partial \boldsymbol{W}^{(2)}}}_{\text{the second layer}}, \underbrace{\frac{\partial \boldsymbol{X}^{(2)}}{\partial \boldsymbol{W}^{(1)}_{(:,1)}}, \cdots, \frac{\partial \boldsymbol{X}^{(2)}}{\partial \boldsymbol{W}^{(1)}_{(:,h_1)}}}_{\text{the first layer}} \right]^\top . \tag{24}$$

Using the chain rule, we can calculate the derivative of $f_\theta(\boldsymbol{G})$ w.r.t. the second-layer weights $\boldsymbol{W}^{(2)}$, which is a vector of size $\kappa_2 h_1$, as follows:

$$\begin{aligned}
\frac{\partial \boldsymbol{X}^{(2)}}{\partial \boldsymbol{W}^{(2)}} &= \frac{\partial \mathbf{1}_n^\top \boldsymbol{A}^{[\kappa_2]} \mathrm{diag}(\boldsymbol{X}^{(1)}; \kappa_2) \cdot \boldsymbol{W}^{(2)}}{\partial \boldsymbol{W}^{(2)}} \\
&= \mathbf{1}_n^\top \boldsymbol{A}^{[\kappa_2]} \mathrm{diag}(\boldsymbol{X}^{(1)}; \kappa_2) \\
&= \mathbf{1}_n^\top \underbrace{\boldsymbol{A}^{[\kappa_2]}}_{\text{size: } n \times \kappa_2 n} \overbrace{\mathrm{diag}(\sigma(\boldsymbol{A}^{[\kappa_1]} \mathrm{diag}(\boldsymbol{X}^{(0)}; \kappa_1) \boldsymbol{W}^{(1)}); \kappa_2))}^{\text{size: } \kappa_2 n \times \kappa_2 h_1} .
\end{aligned} \tag{25}$$
$$\underbrace{\phantom{\mathbf{1}_n^\top \boldsymbol{A}^{[\kappa_2]} \mathrm{diag}(\sigma(\boldsymbol{A}^{[\kappa_1]} \mathrm{diag}(\boldsymbol{X}^{(0)}; \kappa_1) \boldsymbol{W}^{(1)}); \kappa_2))}}_{\text{size: } 1 \times \kappa_2 h_1}$$

The derivative of $f_\theta(\boldsymbol{G})$ w.r.t. the first-layer weights is more complex. For $i \in \mathbb{N}_{h_1}$

$$\frac{\partial \boldsymbol{X}^{(2)}}{\partial \boldsymbol{W}^{(1)}_{(:,i)}} = \frac{\partial \mathbf{1}^\top \boldsymbol{A}^{[\kappa_2]} \mathrm{diag}(\boldsymbol{X}^{(1)} \boldsymbol{e}_i \boldsymbol{e}_i^\top; \kappa_2) \cdot \boldsymbol{W}^{(2)}}{\partial \boldsymbol{W}^{(1)}_{(:,i)}}$$

$$= \frac{\partial \mathbf{1}^\top \boldsymbol{A}^{[\kappa_2]} \overbrace{\mathrm{diag}(\boldsymbol{X}^{(1)} \boldsymbol{e}_i; \kappa_2)}^{\text{size: } \kappa_2 n \times \kappa_2} \cdot \overbrace{\mathrm{diag}(\boldsymbol{e}_i^\top; \kappa_2)}^{\text{size: } \kappa_2 \times \kappa_2 h_1} \boldsymbol{W}^{(2)}}{\partial \boldsymbol{W}^{(1)}_{(:,i)}}$$

$$= \frac{\partial \mathbf{1}^\top \boldsymbol{A}^{[\kappa_2]} \overbrace{\mathrm{diag}(\boldsymbol{X}^{(1)} \boldsymbol{e}_i; \kappa_2)}^{\text{size: } \kappa_2 n \times \kappa_2} \cdot \begin{pmatrix} \overbrace{\boldsymbol{W}^{(2)}_{(i-h_1+h_1)}}^{\text{size: } \kappa_2 \times 1} \\ \cdots \\ \boldsymbol{W}^{(2)}_{(i-h_1+\kappa_2 h_1)} \end{pmatrix}}{\partial \boldsymbol{W}^{(1)}_{(:,i)}}$$

$$= \frac{\partial \mathbf{1}^\top \boldsymbol{A}^{[\kappa_2]} \begin{pmatrix} \overbrace{\boldsymbol{X}^{(1)} \boldsymbol{e}_i \boldsymbol{W}^{(2)}_{(i-h_1+h_1)}}^{\text{size: } \kappa_2 n \times 1} \\ \cdots \\ \boldsymbol{X}^{(1)} \boldsymbol{e}_i \boldsymbol{W}^{(2)}_{(i-h_1+\kappa_2 h_1)} \end{pmatrix}}{\partial \boldsymbol{W}^{(1)}_{(:,i)}}$$

$$= \mathbf{1}^\top \boldsymbol{A}^{[\kappa_2]} \begin{pmatrix} \frac{\partial \boldsymbol{X}^{(1)} \boldsymbol{e}_i}{\partial \boldsymbol{W}^{(1)}_{(:,i)}} \boldsymbol{W}^{(2)}_{(i-h_1+h_1)} \\ \cdots \\ \frac{\partial \boldsymbol{X}^{(1)} \boldsymbol{e}_i}{\partial \boldsymbol{W}^{(1)}_{(:,i)}} \boldsymbol{W}^{(2)}_{(i-h_1+\kappa_2 h_1)} \end{pmatrix}$$

$$= \mathbf{1}^\top \underbrace{\boldsymbol{A}^{[\kappa_2]}}_{\text{size: } n \times \kappa_2 n} \underbrace{\begin{pmatrix} \overbrace{\dot\sigma \cdot \boldsymbol{A}^{[\kappa_1]} \mathrm{diag}(\boldsymbol{X}^{(0)}; \kappa_1) \cdot \boldsymbol{W}^{(2)}_{(i-h_1+h_1)}}^{\text{size: } \kappa_2 n \times \kappa_1 h_0} \\ \cdots \\ \dot\sigma \cdot \boldsymbol{A}^{[\kappa_1]} \mathrm{diag}(\boldsymbol{X}^{(0)}; \kappa_1) \cdot \boldsymbol{W}^{(2)}_{(i-h_1+\kappa_2 h_1)} \end{pmatrix}}_{\text{size: } 1 \times \kappa_1 h_0}, \tag{26}$$

with $\dot\sigma = \frac{\partial \sigma(\boldsymbol{A}^{[\kappa_1]} \mathrm{diag}(\boldsymbol{X}^{(0)}; \kappa_1) \boldsymbol{W}^{(1)}_{(:,i)})}{\partial \boldsymbol{A}^{[\kappa_1]} \mathrm{diag}(\boldsymbol{X}^{(0)}; \kappa_1) \boldsymbol{W}^{(1)}_{(:,i)}}$. Orange marks the first-layer elements, green colors the second-layer elements, and $\mathbf{1}_n^\top$ in purple refers to the summation pooling. If we use ReLU as the activation function, $\dot\sigma \cdot \boldsymbol{A}^{[\kappa_1]} \mathrm{diag}(\boldsymbol{X}^{(0)}; \kappa_1) = \sigma\left(\boldsymbol{A}^{[\kappa_1]} \mathrm{diag}(\boldsymbol{X}^{(0)}; \kappa_1)\right)$.

For the GCN shown in Figure 2, the derivative, *i.e.*, Equation 25 is specified with $\kappa_1, \kappa_2 = 3, 2$ as

$$\frac{\partial \boldsymbol{X}^{(2)}}{\partial \boldsymbol{W}^{(2)}} = \mathbf{1}^\top \underbrace{\boldsymbol{A}^{[2]}}_{\text{size: } n \times 2n} \underbrace{\overbrace{\mathrm{diag}(\sigma(\boldsymbol{A}^{[3]} \mathrm{diag}(\boldsymbol{X}^{(0)}; 3) \boldsymbol{W}^{(1)}); 2)}^{\text{size: } 2n \times 2h_1}}_{\text{size: } 1 \times 2h_1}. \tag{27}$$

$$\frac{\partial \boldsymbol{X}^{(2)}}{\partial \boldsymbol{W}^{(1)}_{(:,i)}} = \mathbf{1}^\top \underbrace{\boldsymbol{A}^{[2]}}_{\text{size: } n \times 2n} \underbrace{\begin{pmatrix} \overbrace{\dot\sigma \cdot \boldsymbol{A}^{[3]} \mathrm{diag}(\boldsymbol{X}^{(0)}; 3) \cdot \boldsymbol{W}^{(2)}_{(i)}}^{\text{size: } 2n \times 3h_0} \\ \dot\sigma \cdot \boldsymbol{A}^{[3]} \mathrm{diag}(\boldsymbol{X}^{(0)}; 3) \cdot \boldsymbol{W}^{(2)}_{(i+h_1)} \end{pmatrix}}_{\text{size: } 1 \times 3h_0}, \tag{28}$$

When a graph is input into a GCN, the adjacency matrix $\boldsymbol{A}$ governs the operations between nodes, ensuring the update is structure-aware by performing row-wise operations on the feature matrix. Meanwhile, the weight matrix $\boldsymbol{W}$ controls how the features are processed, by performing column-wise operations on the feature matrix.

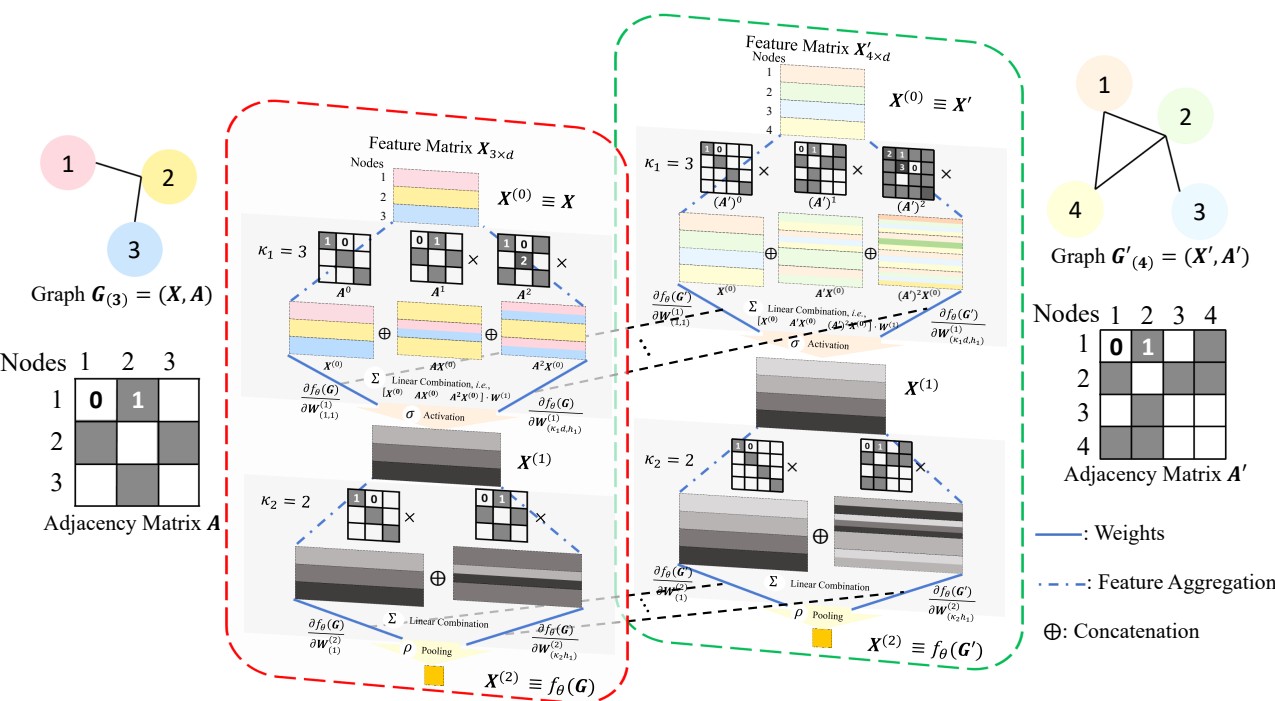

Figure 5: Graphical illustration of GNTK computation: $K_\theta(\boldsymbol{G}_{(3)}, \boldsymbol{G}'_{(4)}) = \left\langle \frac{\partial f_\theta(\boldsymbol{G})}{\partial \theta}, \frac{\partial f_\theta(\boldsymbol{G}')}{\partial \theta} \right\rangle = \frac{\partial f_\theta(\boldsymbol{G})}{\partial \boldsymbol{W}^{(1)}_{(1,1)}} \frac{\partial f_\theta(\boldsymbol{G}')}{\partial \boldsymbol{W}^{(1)}_{(1,1)}} + \cdots +$
$\frac{\partial f_\theta(\boldsymbol{G})}{\partial \boldsymbol{W}^{(1)}_{(\kappa_1 d, h_1)}} \frac{\partial f_\theta(\boldsymbol{G}')}{\partial \boldsymbol{W}^{(1)}_{(\kappa_1 d, h_1)}} + \frac{\partial f_\theta(\boldsymbol{G})}{\partial \boldsymbol{W}^{(2)}_{(1)}} \frac{\partial f_\theta(\boldsymbol{G}')}{\partial \boldsymbol{W}^{(2)}_{(1)}} + \cdots + \frac{\partial f_\theta(\boldsymbol{G})}{\partial \boldsymbol{W}^{(2)}_{(\kappa_2 h_1)}} \frac{\partial f_\theta(\boldsymbol{G}')}{\partial \boldsymbol{W}^{(2)}_{(\kappa_2 h_1)}}.$

## A.3. Graph Neural Tangent Kernel (GNTK)

By substituting the parameter evolution (Equation 10)

$$\frac{\partial \theta^t}{\partial t} = -\frac{\eta}{N} \left[ \frac{\partial \mathcal{L}(f_{\theta^t}(\boldsymbol{G}_i), \boldsymbol{y}_i)}{\partial f_{\theta^t}(\boldsymbol{G}_i)} \right]_N^\top \cdot \left[ \frac{\partial f_{\theta^t}(\boldsymbol{G}_i)}{\partial \theta^t} \right]_N. \tag{29}$$

into the first-order approximation term $(*)$ of Equation 16, it obtains

$$
\begin{aligned}
(*) &= \left\langle \frac{\partial f_{\theta^t}(\cdot)}{\partial \theta^t}, -\frac{\eta}{N} \left[ \frac{\partial \mathcal{L}(f_{\theta^t}(\boldsymbol{G}_i), \boldsymbol{y}_i)}{\partial f_{\theta^t}(\boldsymbol{G}_i)} \right]_N^\top \cdot \left[ \frac{\partial f_{\theta^t}(\boldsymbol{G}_i)}{\partial \theta^t} \right]_N \right\rangle \\
&= -\frac{\eta}{N} \left[ \frac{\partial \mathcal{L}(f_{\theta^t}(\boldsymbol{G}_i), \boldsymbol{y}_i)}{\partial f_{\theta^t}(\boldsymbol{G}_i)} \right]_N^\top \cdot \left\langle \frac{\partial f_{\theta^t}(\cdot)}{\partial \theta^t}, \left[ \frac{\partial f_{\theta^t}(\boldsymbol{G}_i)}{\partial \theta^t} \right]_N \right\rangle \\
&= -\frac{\eta}{N} \left[ \frac{\partial \mathcal{L}(f_{\theta^t}(\boldsymbol{G}_i), \boldsymbol{y}_i)}{\partial f_{\theta^t}(\boldsymbol{G}_i)} \right]_N^\top \cdot \left[ \left\langle \frac{\partial f_{\theta^t}(\cdot)}{\partial \theta^t}, \frac{\partial f_{\theta^t}(\boldsymbol{G}_i)}{\partial \theta^t} \right\rangle \right]_N \\
&= -\frac{\eta}{N} \left[ \frac{\partial \mathcal{L}(f_{\theta^t}(\boldsymbol{G}_i), \boldsymbol{y}_i)}{\partial f_{\theta^t}(\boldsymbol{G}_i)} \right]_N^\top \cdot \left[ K_{\theta^t}(\boldsymbol{G}_i, \cdot) \right]_N, \tag{30}
\end{aligned}
$$

which derives Equation 17 as

$$\frac{\partial f_{\theta^t}}{\partial t} = -\frac{\eta}{N} \left[ \frac{\partial \mathcal{L}(f_{\theta^t}(\boldsymbol{G}_i), \boldsymbol{y}_i)}{\partial f_{\theta^t}(\boldsymbol{G}_i)} \right]_N^\top \cdot \left[ K_{\theta^t}(\boldsymbol{G}_i, \cdot) \right]_N + o\left( \frac{\partial \theta^t}{\partial t} \right), \tag{31}$$

where the symmetric and positive definite $K_{\theta^t}(\boldsymbol{G}_i, \cdot) \coloneqq \left\langle \frac{\partial f_{\theta^t}(\boldsymbol{G}_i)}{\partial \theta^t}, \frac{\partial f_{\theta^t}(\cdot)}{\partial \theta^t} \right\rangle$ is referred to as graph neural tangent kernel (GNTK) (Du et al., 2019; Krishnagopal & Ruiz, 2023). Figure 5 illustrates the GNTK calculation process. In simple terms,

examining a model's behavior by focusing on the model itself, rather than its parameters, often involves the use of kernel functions.

It can be observed that the quantity $\frac{\partial f_{\theta^t}(\cdot)}{\partial \theta^t}$ representing the partial derivative of the GCN with respect to its parameters, present in $K_{\theta^t}(\boldsymbol{G}_i, \cdot) = \left\langle \frac{\partial f_{\theta^t}(\boldsymbol{G}_i)}{\partial \theta^t}, \frac{\partial f_{\theta^t}(\cdot)}{\partial \theta^t} \right\rangle$, is determined by both the structure and the specific parameters $\theta^t$, but does not depend on the input graphs. The other term $\frac{\partial f_{\theta^t}(\boldsymbol{G}_i)}{\partial \theta^t}$ relies not only on the GCN structure and specific $\theta^t$, but also on the input graph. If the input for $\frac{\partial f_{\theta^t}(\boldsymbol{G}_i)}{\partial \theta^t}$ is not specified, the GNTK simplifies to a general form $K_{\theta^t}(\cdot, \cdot)$. When a specific graph $\boldsymbol{G}_j$ is defined as the input for $\frac{\partial f_{\theta^t}(\cdot)}{\partial \theta^t}$, GNTK becomes a scalar as $K_{\theta^t}(\boldsymbol{G}_i, \boldsymbol{G}_j) = \langle \frac{\partial f_{\theta^t}(\boldsymbol{G}_i)}{\partial \theta^t}, \frac{\partial f_{\theta^t}(\boldsymbol{G}_j)}{\partial \theta^t} \rangle$. These are in line with the kernel used in functional gradient descent. By providing the input graph $\boldsymbol{G}_i$, one coordinate of $K_{\theta^t}$ is fixed, causing the GCN to update along $K_{\theta^t}(\boldsymbol{G}_i, \cdot)$, based on the magnitude of $\frac{\partial f_{\theta^t}(\boldsymbol{G}_i)}{\partial \theta^t}$. This process aligns with the core principle of functional gradient descent. In summary, both the GNTK and the canonical kernel are consistent in their mathematical formulation and show alignment in how they affect the evolution of the corresponding GCN. Furthermore, Theorem 5 highlights the asymptotic connection between the GNTK and the canonical kernel used in functional gradient descent.

## B. Detailed Proofs

Before providing the detailed proofs, we first introduce the gradient of an evaluation functional $E_{\boldsymbol{G}}(f)$.

**Lemma 7.** *The gradient of an evaluation functional $E_{\boldsymbol{G}}(f) = f(\boldsymbol{G}) : \mathcal{H} \mapsto \mathbb{R}$ is $\nabla_f E_{\boldsymbol{G}}(f) = K_{\boldsymbol{G}}$.*

**Proof of Lemma 7**  Let us define a function $\phi$ by adding a small perturbation $\epsilon g$ ($\epsilon \in \mathbb{R}, g \in \mathcal{H}$) to $f \in \mathcal{H}$, so that $\phi = f + \epsilon g$. $\phi \in \mathcal{H}$ since RKHS is closed under addition and scalar multiplication. Therefore, for an evaluation functional $E_{\boldsymbol{G}}[f] = f(\boldsymbol{G}) : \mathcal{H} \mapsto \mathbb{R}$, we can evaluate $\phi$ at $\boldsymbol{G}$ as

$$
\begin{aligned}
E_{\boldsymbol{G}}[\phi] &= E_{\boldsymbol{G}}[f + \epsilon g] \\
&= E_{\boldsymbol{G}}[f] + \epsilon E_{\boldsymbol{G}}[g] + 0 \\
&= E_{\boldsymbol{G}}[f] + \epsilon \langle K(\boldsymbol{G}, \cdot), g \rangle_{\mathcal{H}} + 0
\end{aligned}
\tag{32}
$$

Recall implicit definition of Fréchet derivative in RKHS (see Definition 2) $E_{\boldsymbol{G}}[f + \epsilon g] = E_{\boldsymbol{G}}[f] + \epsilon \langle \nabla_f E_{\boldsymbol{G}}[f], g \rangle_{\mathcal{H}} + o(\epsilon)$, it follows from Equation 32 that we have the gradient of a evaluation functional $\nabla_f E_{\boldsymbol{G}}[f] = K_{\boldsymbol{G}}$.

∎

**Proof of Theorem 5** By describing the evolution of a GCN in terms of parameter variations and from a high-level perspective within the function space, we obtain

$$
-\frac{\eta}{N} \left[ \frac{\partial \mathcal{L}(f_{\theta^t}(\boldsymbol{G}_i), \boldsymbol{y}_i)}{\partial f_{\theta^t}(\boldsymbol{G}_i)} \right]_N^\top \cdot [K(\boldsymbol{G}_i, \cdot)]_N = -\frac{\eta}{N} \left[ \frac{\partial \mathcal{L}(f_{\theta^t}(\boldsymbol{G}_i), \boldsymbol{y}_i)}{\partial f_{\theta^t}(\boldsymbol{G}_i)} \right]_N^\top \cdot \left[ \left\langle \frac{\partial f_{\theta^t}(\boldsymbol{G}_i)}{\partial \theta^t}, \frac{\partial f_{\theta^t}(\cdot)}{\partial \theta^t} \right\rangle \right]_N + o\left( \frac{\partial \theta^t}{\partial t} \right).
\tag{33}
$$

After the reorganization, we get

$$
-\frac{\eta}{N} \left[ \frac{\partial \mathcal{L}(f_{\theta^t}(\boldsymbol{G}_i), \boldsymbol{y}_i)}{\partial f_{\theta^t}(\boldsymbol{G}_i)} \right]_N^\top \cdot [K(\boldsymbol{G}_i, \cdot) - K_{\theta^t}(\boldsymbol{G}_i, \cdot)]_N = o\left( \frac{\partial \theta^t}{\partial t} \right).
\tag{34}
$$

By inserting the evolution of the parameters

$$
\frac{\partial \theta^t}{\partial t} = -\eta \frac{\partial \mathcal{L}}{\partial \theta^t} = -\frac{\eta}{N} \left[ \frac{\partial \mathcal{L}(f_{\theta^t}(\boldsymbol{G}_i), \boldsymbol{y}_i)}{\partial f_{\theta^t}(\boldsymbol{G}_i)} \right]_N^\top \cdot \left[ \frac{\partial f_{\theta^t}(\boldsymbol{G}_i)}{\partial \theta^t} \right]_N
\tag{35}
$$

into the remainder, we have

$$
-\frac{\eta}{N} \left[ \frac{\partial \mathcal{L}(f_{\theta^t}(\boldsymbol{G}_i), \boldsymbol{y}_i)}{\partial f_{\theta^t}(\boldsymbol{G}_i)} \right]_N^\top \cdot [K(\boldsymbol{G}_i, \cdot) - K_{\theta^t}(\boldsymbol{G}_i, \cdot)]_N = o\left( -\frac{\eta}{N} \left[ \frac{\partial \mathcal{L}(f_{\theta^t}(\boldsymbol{G}_i), \boldsymbol{y}_i)}{\partial f_{\theta^t}(\boldsymbol{G}_i)} \right]_N^\top \cdot \left[ \frac{\partial f_{\theta^t}(\boldsymbol{G}_i)}{\partial \theta^t} \right]_N \right).
\tag{36}
$$

When training a GCN with a convex loss $\mathcal{L}$, which is convex concerning $f_\theta$ but often not with regard to $\theta$, we have the limit of the vector $\lim_{t \to \infty} \left[ \frac{\partial \mathcal{L}(f_{\theta^t}(\boldsymbol{G}_i), \boldsymbol{y}_i)}{\partial f_{\theta^t}(\boldsymbol{G}_i)} \right]_N = \boldsymbol{0}$. Since the right side of the equation is a higher order infinitesimal than the left, to preserve this equality, it leads us to the conclusion that

$$
\lim_{t \to \infty} [K(\boldsymbol{G}_i, \cdot) - K_{\theta^t}(\boldsymbol{G}_i, \cdot)]_N = \boldsymbol{0}.
\tag{37}
$$

This means that for each $\boldsymbol{G} \in \{\boldsymbol{G}_i\}_N$, GNTK converges pointwise to the canonical kernel.

∎

**Proof of Proposition 6** By recalling the definition of the Fréchet derivative in Definition 2, the convexity of $\mathcal{L}$ implies that

$$
\frac{\partial \mathcal{L}}{\partial t} \le \underbrace{\left\langle \frac{\partial \mathcal{L}}{\partial f_{\theta^{t+1}}}, \frac{f_{\theta^t}}{\partial t} \right\rangle_{\mathcal{H}}}_{\Xi}.
\tag{38}
$$

By identifying the Fréchet derivative of $\frac{\partial \mathcal{L}}{\partial f_{\theta^{t+1}}}$ and the evolution of $f_{\theta^t}$, the right-hand side term $\Xi$ can be represented as

$$
\begin{aligned}
\Xi &= \left\langle \mathcal{G}^{t+1}, -\eta \mathcal{G}^t \right\rangle_{\mathcal{H}} \\
&= -\frac{\eta}{N^2} \left\langle \left[ \frac{\partial \mathcal{L}(f_{\theta^{t+1}}(\boldsymbol{G}_i), \boldsymbol{y}_i)}{\partial f_{\theta^{t+1}}(\boldsymbol{G}_i)} \right]_N^\top \cdot [K_{\boldsymbol{G}_i}]_N, [K_{\boldsymbol{G}_i}]_N^\top \cdot \left[ \frac{\partial \mathcal{L}(f_{\theta^t}(\boldsymbol{G}_i), \boldsymbol{y}_i)}{\partial f_{\theta^t}(\boldsymbol{G}_i)} \right]_N \right\rangle_{\mathcal{H}} \\
&= -\frac{\eta}{N^2} \left[ \frac{\partial \mathcal{L}(f_{\theta^{t+1}}(\boldsymbol{G}_i), \boldsymbol{y}_i)}{\partial f_{\theta^{t+1}}(\boldsymbol{G}_i)} \right]_N^\top \cdot \left\langle [K_{\boldsymbol{G}_i}]_N, [K_{\boldsymbol{G}_i}]_N^\top \right\rangle_{\mathcal{H}} \cdot \left[ \frac{\partial \mathcal{L}(f_{\theta^t}(\boldsymbol{G}_i), \boldsymbol{y}_i)}{\partial f_{\theta^t}(\boldsymbol{G}_i)} \right]_N \\
&= -\frac{\eta}{N} \left[ \frac{\partial \mathcal{L}(f_{\theta^t}(\boldsymbol{G}_i), \boldsymbol{y}_i)}{\partial f_{\theta^t}(\boldsymbol{G}_i)} \right]_N^\top \bar{\boldsymbol{K}} \left[ \frac{\partial \mathcal{L}(f_{\theta^{t+1}}(\boldsymbol{G}_i), \boldsymbol{y}_i)}{\partial f_{\theta^{t+1}}(\boldsymbol{G}_i)} \right]_N,
\end{aligned}
\tag{39}
$$

where $\bar{\boldsymbol{K}} = \boldsymbol{K}/N$, and $\boldsymbol{K}$ is an $N \times N$ symmetric, positive definite matrix with elements $K(\boldsymbol{G}_i, \boldsymbol{G}_j)$ positioned at the $i$-th row and $j$-th column. For convenience, we adopt the simplified notation that $\frac{\partial \mathcal{L}(f_{\theta\square}(\boldsymbol{G}_i), \boldsymbol{y}_i)}{\partial f_{\theta\square}(\boldsymbol{G}_i)} := \partial_{f_{\theta\square}} \mathcal{L}(f_{\theta\square}; \boldsymbol{G}_i)$. The final term in Equation 39 can be rewritten as

$$
\begin{aligned}
&-\frac{\eta}{N} \left[ \partial_{f_{\theta^t}} \mathcal{L}(f_{\theta^t}; \boldsymbol{G}_i) \right]_N^\top \bar{\boldsymbol{K}} \left[ \partial_{f_{\theta^{t+1}}} \mathcal{L}(f_{\theta^{t+1}}; \boldsymbol{G}_i) \right]_N \\
=~ &-\frac{\eta}{N} \left[ \partial_{f_{\theta^t}} \mathcal{L}(f_{\theta^t}; \boldsymbol{G}_i) \right]_N^\top \bar{\boldsymbol{K}} \left( \left[ \partial_{f_{\theta^{t+1}}} \mathcal{L}(f_{\theta^{t+1}}; \boldsymbol{G}_i) \right]_N + \left[ \partial_{f_{\theta^t}} \mathcal{L}(f_{\theta^t}; \boldsymbol{G}_i) \right]_N - \left[ \partial_{f_{\theta^t}} \mathcal{L}(f_{\theta^t}; \boldsymbol{G}_i) \right]_N \right) \\
=~ &-\frac{\eta}{N} \left[ \partial_{f_{\theta^t}} \mathcal{L}(f_{\theta^t}; \boldsymbol{G}_i) \right]_N^\top \bar{\boldsymbol{K}} \left[ \partial_{f_{\theta^t}} \mathcal{L}(f_{\theta^t}; \boldsymbol{G}_i) \right]_N \\
&-\frac{\eta}{N} \left[ \partial_{f_{\theta^t}} \mathcal{L}(f_{\theta^t}; \boldsymbol{G}_i) \right]_N^\top \bar{\boldsymbol{K}} \left( \left[ \partial_{f_{\theta^{t+1}}} \mathcal{L}(f_{\theta^{t+1}}; \boldsymbol{G}_i) \right]_N - \left[ \partial_{f_{\theta^t}} \mathcal{L}(f_{\theta^t}; \boldsymbol{G}_i) \right]_N \right) \\
=~ &-\frac{\eta}{N} \left[ \partial_{f_{\theta^t}} \mathcal{L}(f_{\theta^t}; \boldsymbol{G}_i) \right]_N^\top \bar{\boldsymbol{K}} \left[ \partial_{f_{\theta^t}} \mathcal{L}(f_{\theta^t}; \boldsymbol{G}_i) \right]_N \\
&+\frac{\eta}{N} \left( \left[ \partial_{f_{\theta^{t+1}}} \mathcal{L}(f_{\theta^{t+1}}; \boldsymbol{G}_i) \right]_N^\top - \left[ \partial_{f_{\theta^t}} \mathcal{L}(f_{\theta^t}; \boldsymbol{G}_i) \right]_N^\top - \left[ \partial_{f_{\theta^{t+1}}} \mathcal{L}(f_{\theta^{t+1}}; \boldsymbol{G}_i) \right]_N^\top \right) \\
&\cdot \bar{\boldsymbol{K}} \cdot \left( \left[ \partial_{f_{\theta^{t+1}}} \mathcal{L}(f_{\theta^{t+1}}; \boldsymbol{G}_i) \right]_N - \left[ \partial_{f_{\theta^t}} \mathcal{L}(f_{\theta^t}; \boldsymbol{G}_i) \right]_N \right).
\end{aligned}
\tag{40}
$$

The last term in Equation 40 above can be further detailed as

$$
\begin{aligned}
&\frac{\eta}{N} \left( \left[ \partial_{f_{\theta^{t+1}}} \mathcal{L}(f_{\theta^{t+1}}; \boldsymbol{G}_i) \right]_N^\top - \left[ \partial_{f_{\theta^t}} \mathcal{L}(f_{\theta^t}; \boldsymbol{G}_i) \right]_N^\top - \left[ \partial_{f_{\theta^{t+1}}} \mathcal{L}(f_{\theta^{t+1}}; \boldsymbol{G}_i) \right]_N^\top \right) \\
&\cdot \bar{\boldsymbol{K}} \left( \left[ \partial_{f_{\theta^{t+1}}} \mathcal{L}(f_{\theta^{t+1}}; \boldsymbol{G}_i) \right]_N - \left[ \partial_{f_{\theta^t}} \mathcal{L}(f_{\theta^t}; \boldsymbol{G}_i) \right]_N \right) \\
=~ &\frac{\eta}{N} \left( \left[ \partial_{f_{\theta^{t+1}}} \mathcal{L}(f_{\theta^{t+1}}; \boldsymbol{G}_i) \right]_N - \left[ \partial_{f_{\theta^t}} \mathcal{L}(f_{\theta^t}; \boldsymbol{G}_i) \right]_N \right)^\top \bar{\boldsymbol{K}} \left( \left[ \partial_{f_{\theta^{t+1}}} \mathcal{L}(f_{\theta^{t+1}}; \boldsymbol{G}_i) \right]_N - \left[ \partial_{f_{\theta^t}} \mathcal{L}(f_{\theta^t}; \boldsymbol{G}_i) \right]_N \right) \\
&-\frac{\eta}{N} \left[ \partial_{f_{\theta^{t+1}}} \mathcal{L}(f_{\theta^{t+1}}; \boldsymbol{G}_i) \right]_N^\top \bar{\boldsymbol{K}} \left( \left[ \partial_{f_{\theta^{t+1}}} \mathcal{L}(f_{\theta^{t+1}}; \boldsymbol{G}_i) \right]_N - \left[ \partial_{f_{\theta^t}} \mathcal{L}(f_{\theta^t}; \boldsymbol{G}_i) \right]_N \right) \\
=~ &\frac{\eta}{N} \left[ \partial_{f_{\theta^{t+1}}} \mathcal{L}(f_{\theta^{t+1}}; \boldsymbol{G}_i) - \partial_{f_{\theta^t}} \mathcal{L}(f_{\theta^t}; \boldsymbol{G}_i) \right]_N^\top \bar{\boldsymbol{K}} \left[ \partial_{f_{\theta^{t+1}}} \mathcal{L}(f_{\theta^{t+1}}; \boldsymbol{G}_i) - \partial_{f_{\theta^t}} \mathcal{L}(f_{\theta^t}; \boldsymbol{G}_i) \right]_N \\
&-\frac{\eta}{N} \left( \left[ \partial_{f_{\theta^{t+1}}} \mathcal{L}(f_{\theta^{t+1}}; \boldsymbol{G}_i) \right]_N - \frac{1}{2} \left[ \partial_{f_{\theta^t}} \mathcal{L}(f_{\theta^t}; \boldsymbol{G}_i) \right]_N \right)^\top \bar{\boldsymbol{K}} \left( \left[ \partial_{f_{\theta^{t+1}}} \mathcal{L}(f_{\theta^{t+1}}; \boldsymbol{G}_i) \right]_N - \frac{1}{2} \left[ \partial_{f_{\theta^t}} \mathcal{L}(f_{\theta^t}; \boldsymbol{G}_i) \right]_N \right) \\
&+\frac{\eta}{4N} \left[ \partial_{f_{\theta^t}} \mathcal{L}(f_{\theta^t}; \boldsymbol{G}_i) \right]_N^\top \bar{\boldsymbol{K}} \left[ \partial_{f_{\theta^t}} \mathcal{L}(f_{\theta^t}; \boldsymbol{G}_i) \right]_N.
\end{aligned}
\tag{41}
$$

Since $\bar{\boldsymbol{K}}$ is positive definite, it is apparent that

$$
\frac{\eta}{N} \left( \left[ \partial_{f_{\theta^{t+1}}} \mathcal{L}(f_{\theta^{t+1}}; \boldsymbol{G}_i) \right]_N - \frac{1}{2} \left[ \partial_{f_{\theta^t}} \mathcal{L}(f_{\theta^t}; \boldsymbol{G}_i) \right]_N \right)^\top \bar{\boldsymbol{K}} \left( \left[ \partial_{f_{\theta^{t+1}}} \mathcal{L}(f_{\theta^{t+1}}; \boldsymbol{G}_i) \right]_N - \frac{1}{2} \left[ \partial_{f_{\theta^t}} \mathcal{L}(f_{\theta^t}; \boldsymbol{G}_i) \right]_N \right)
$$

is a non-negative term. Hence, by combining Equations 39, 40, and 41, we obtain

$$
\begin{aligned}
\Xi \leq~ &-\frac{3\eta}{4N} \underbrace{\left[ \partial_{f_{\theta^t}} \mathcal{L}(f_{\theta^t}; \boldsymbol{G}_i) \right]_N^\top \bar{\boldsymbol{K}} \left[ \partial_{f_{\theta^t}} \mathcal{L}(f_{\theta^t}; \boldsymbol{G}_i) \right]_N}_{\text{①}} \\
&+\frac{\eta}{N} \underbrace{\left[ \partial_{f_{\theta^{t+1}}} \mathcal{L}(f_{\theta^{t+1}}; \boldsymbol{G}_i) - \partial_{f_{\theta^t}} \mathcal{L}(f_{\theta^t}; \boldsymbol{G}_i) \right]_N^\top \bar{\boldsymbol{K}} \left[ \partial_{f_{\theta^{t+1}}} \mathcal{L}(f_{\theta^{t+1}}; \boldsymbol{G}_i) - \partial_{f_{\theta^t}} \mathcal{L}(f_{\theta^t}; \boldsymbol{G}_i) \right]_N}_{\text{②}}.
\end{aligned}
\tag{42}
$$

Based on the definition of the evaluation functional and the assumption that $\mathcal{L}$ is Lipschitz smooth with a constant $\tau > 0$, the term ② in the final part of Equation 42 is bounded above as

$$
\begin{aligned}
② \quad &= \quad \left[\partial_{f_{\theta^{t+1}}}\mathcal{L}(f_{\theta^{t+1}}; \boldsymbol{G}_i) - \partial_{f_{\theta^t}}\mathcal{L}(f_{\theta^t}; \boldsymbol{G}_i)\right]_N^\top \bar{\boldsymbol{K}} \left[\partial_{f_{\theta^{t+1}}}\mathcal{L}(f_{\theta^{t+1}}; \boldsymbol{G}_i) - \partial_{f_{\theta^t}}\mathcal{L}(f_{\theta^t}; \boldsymbol{G}_i)\right]_N \\
&= \quad \left[E_{\boldsymbol{G}_i}\left(\frac{\partial\mathcal{L}(f_{\theta^{t+1}})}{\partial f_{\theta^{t+1}}} - \frac{\partial\mathcal{L}(f_{\theta^t})}{\partial f_{\theta^t}}\right)\right]_N^\top \bar{\boldsymbol{K}}\left[E_{\boldsymbol{G}_i}\left(\frac{\partial\mathcal{L}(f_{\theta^{t+1}})}{\partial f_{\theta^{t+1}}} - \frac{\partial\mathcal{L}(f_{\theta^t})}{\partial f_{\theta^t}}\right)\right]_N \\
&\leq \quad \tau^2\left[E_{\boldsymbol{G}_i}\left(f_{\theta^{t+1}} - f_{\theta^t}\right)\right]_N^\top \bar{\boldsymbol{K}}\left[E_{\boldsymbol{G}_i}\left(f_{\theta^{t+1}} - f_{\theta^t}\right)\right]_N \\
&= \quad \tau^2\left\langle\left(f_{\theta^{t+1}} - f_{\theta^t}\right), [K_{\boldsymbol{G}_i}]_N^\top\right\rangle_{\mathcal{H}} \cdot \bar{\boldsymbol{K}} \cdot \left\langle[K_{\boldsymbol{G}_i}]_N, \left(f_{\theta^{t+1}} - f_{\theta^t}\right)\right\rangle_{\mathcal{H}} \\
&= \quad \eta^2\tau^2 \cdot \left[\partial_{f_{\theta^t}}\mathcal{L}(f_{\theta^t}; \boldsymbol{G}_i)\right]_N^\top \frac{\left\langle[K_{\boldsymbol{G}_i}]_N, [K_{\boldsymbol{G}_i}]_N^\top\right\rangle_{\mathcal{H}}}{N} \cdot \bar{\boldsymbol{K}} \cdot \frac{\left\langle[K_{\boldsymbol{G}_i}]_N, [K_{\boldsymbol{G}_i}]_N^\top\right\rangle_{\mathcal{H}}}{N} \cdot \left[\partial_{f_{\theta^t}}\mathcal{L}(f_{\theta^t}; \boldsymbol{G}_i)\right]_N. \quad (43)
\end{aligned}
$$

Under the assumption that the canonical kernel is bounded above by a constant $\gamma > 0$, we have

$$
\left\langle[K_{\boldsymbol{G}_i}]_N, [K_{\boldsymbol{G}_i}]_N^\top\right\rangle_{\mathcal{H}} \leq \gamma\left\langle[1]_N, [1]_N^\top\right\rangle,
$$

and

$$
\bar{\boldsymbol{K}} \leq \frac{\gamma}{N}\left\langle[1]_N, [1]_N^\top\right\rangle.
$$

As a result, ① is bounded above by

$$
\begin{aligned}
① \quad &\leq \quad \frac{\gamma}{N}\left\langle\left[\sum_{i=1}^N \partial_{f_{\theta^t}}\mathcal{L}(f_{\theta^t}; \boldsymbol{G}_i)\right]_N^\top, [1]_N\right\rangle\left\langle[1]_N^\top, \left[\sum_{i=1}^N \partial_{f_{\theta^t}}\mathcal{L}(f_{\theta^t}; \boldsymbol{G}_i)\right]_N\right\rangle \\
&= \quad \frac{\gamma}{N}\left(\sum_{i=1}^N \partial_{f_{\theta^t}}\mathcal{L}(f_{\theta^t}; \boldsymbol{G}_i)\right)^2. \quad (44)
\end{aligned}
$$

Meanwhile, the final term in Equation 43 is also bounded above:

$$
\begin{aligned}
&\eta^2\tau^2 \cdot \left[\partial_{f_{\theta^t}}\mathcal{L}(f_{\theta^t}; \boldsymbol{G}_i)\right]_N^\top \frac{\left\langle[K_{\boldsymbol{G}_i}]_N, [K_{\boldsymbol{G}_i}]_N^\top\right\rangle_{\mathcal{H}}}{N} \cdot \bar{\boldsymbol{K}} \cdot \frac{\left\langle[K_{\boldsymbol{G}_i}]_N, [K_{\boldsymbol{G}_i}]_N^\top\right\rangle_{\mathcal{H}}}{N} \cdot \left[\partial_{f_{\theta^t}}\mathcal{L}(f_{\theta^t}; \boldsymbol{G}_i)\right]_N \\
&\leq \quad \eta^2\tau^2\left[\frac{\gamma}{N}\sum_{i=1}^N \partial_{f_{\theta^t}}\mathcal{L}(f_{\theta^t}; \boldsymbol{G}_i)\right]^\top \cdot \bar{\boldsymbol{K}} \cdot \left[\frac{\gamma}{N}\sum_{i=1}^N \partial_{f_{\theta^t}}\mathcal{L}(f_{\theta^t}; \boldsymbol{G}_i)\right]_N \\
&\leq \quad \frac{\eta^2\tau^2\gamma^3}{N}\left\langle\left[\frac{1}{N}\sum_{i=1}^N \partial_{f_{\theta^t}}\mathcal{L}(f_{\theta^t}; \boldsymbol{G}_i)\right]_N^\top, [1]_N\right\rangle\left\langle[1]_N^\top, \left[\frac{1}{N}\sum_{i=1}^N \partial_{f_{\theta^t}}\mathcal{L}(f_{\theta^t}; \boldsymbol{G}_i)\right]_N\right\rangle \\
&= \quad \frac{\eta^2\tau^2\gamma^3}{N}\left(\sum_{i=1}^N \partial_{f_{\theta^t}}\mathcal{L}(f_{\theta^t}; \boldsymbol{G}_i)\right)^2. \quad (45)
\end{aligned}
$$

Hence, by combining Equations 42, 43, 44, and 45, we derive

$$
\Xi \leq -\eta\gamma\left(\frac{3}{4} - \eta^2\tau^2\gamma^2\right)\left(\frac{1}{N}\sum_{i=1}^N \partial_{f_{\theta^t}}\mathcal{L}(f_{\theta^t}; \boldsymbol{G}_i)\right)^2, \quad (46)
$$

which means

$$
\frac{\partial\mathcal{L}}{\partial t} \leq \Xi \leq -\eta\gamma\left(\frac{3}{4} - \eta^2\tau^2\gamma^2\right)\left(\frac{1}{N}\sum_{i=1}^N \partial_{f_{\theta^t}}\mathcal{L}(f_{\theta^t}; \boldsymbol{G}_i)\right)^2. \quad (47)
$$

Therefore, if $\eta \leq \frac{1}{2\tau\gamma}$, it follows that

$$
\frac{\partial\mathcal{L}}{\partial t} \leq -\frac{\eta\gamma}{2}\left(\frac{1}{N}\sum_{i=1}^N \partial_{f_{\theta^t}}\mathcal{L}(f_{\theta^t}; \boldsymbol{G}_i)\right)^2 = -\frac{\eta\gamma}{2}\left(\frac{1}{N}\sum_{i=1}^N \frac{\partial\mathcal{L}(f_{\theta^t}(\boldsymbol{G}_i), \boldsymbol{y}_i)}{\partial f_{\theta^t}(\boldsymbol{G}_i)}\right)^2. \quad (48)
$$

∎

# C. Experiment Details

This section outlines the experiment details, covering the experimental setup, supplementary results, and a brief analysis of graph-level and node-level tasks on benchmark datasets.

## C.1. Experimental Setup

**Device Setup.** We mainly conduct experiments using NVIDIA Geforce RTX 3090 (24G).

**Dataset Splitting.** The train / val / test split configurations for the benchmark datasets are provided in Table 5.

| Dataset | train | validation | test |
|---|---|---|---|
| QM9 | 110000 | 10000 | 10831 |
| ZINC | 220011 | 24445 | 5000 |
| ogbg-molhiv | 32901 | 4113 | 4113 |
| ogbg-molpcba | 350343 | 43793 | 43793 |
| gen-reg | 30000 | 10000 | 10000 |
| gen-cls | 30000 | 10000 | 10000 |

Table 5: Dataset splitting for the benchmark datasets.

**Hyperparameter Settings.** The key hyperparameter settings for all benchmark datasets are listed in Table 6. The analysis of the hyperparameter, start-ratio, under GraNT(B) is provided in Table 7.

| Dataset | lr | $\kappa$-list | batch-size | start-ratio (GraNT) | epochs |
|---|---|---|---|---|---|
| QM9 | 0.00005 | [3, 2] | 256 | 0.05 | 750 |
| ZINC | 0.0004 | [5, 4, 2, 2] | 256 | 0.05 | 1000 |
| ogbg-molhiv | 0.01 | [4, 3, 2, 2] | 500 | 0.1 | 600 |
| ogbg-molpcba | 0.015 | [5, 4, 3, 2, 2] | 128 | 0.1 | 800 |
| gen-reg | 0.0002 | [3, 2] | 100 | 0.05 | 250 |
| gen-cls | 0.0002 | [4, 3] | 200 | 0.05 | 500 |

Table 6: Key hyperparameter settings for the benchmark datasets, with the "start-ratio" specified for GraNT.

| Dataset | Metric | 0.05 | 0.1 | 0.2 | 0.4 | 0.8 | full |
|---|---|---|---|---|---|---|---|
| QM9 | MAE | 0.0051 | 0.0053 | 0.0053 | 0.0053 | 0.0053 | 0.0051 |
| | Training time (s) | 6392.26 | 6974.30 | 7918.51 | 10828.18 | 14081.66 | 9654.81 |
| ogbg-molhiv | ROC-AUC | 0.7546 | 0.7676 | 0.7652 | 0.7618 | 0.7592 | 0.7572 |
| | Training time (s) | 1362.09 | 1457.39 | 1719.43 | 2157.05 | 3173.92 | 2163.5 |
| gen-cls | ROC-AUC | 0.9157 | 0.9156 | 0.9156 | 0.9156 | 0.9156 | 0.9150 |
| | Training time (s) | 6145.72 | 6237.92 | 6939.95 | 9459.22 | 13153.81 | 11662.25 |

Table 7: Performance comparison *w.r.t.* "start-ratio" for different datasets.

## C.2. Graph-level Tasks

We train the GCN using GraNT (B), GraNT (S), and the "without GraNT", all under a common experimental setup for graph-level tasks. For GraNT (B) and GraNT (S), we adopt a *curriculum learning* strategy (Bengio et al., 2009; Zhang et al., 2024a). Intuitively, at the start of training, the model is undertrained and undergoes significant changes, which calls for more frequent selection by the teacher but with small subsets to help the learner better *digest* the provided graphs; in contrast, by the end of training, the model stabilizes and is able to *digest* large subsets. Specifically, the selection interval progressively widens over 50 stages, beginning from the first epoch and gradually extending to the maximum interval. The initial selection ratios are predefined (*i.e.*, the "start-ratio" hyperparameter) for each benchmark dataset. Additionally, for ogbg-molhiv and ogbg-molpcba, we use the `ReduceLROnPlateau` (Hinton et al., 2012) as the learning rate scheduler, with the 'lr' values in Table 6 representing the initial learning rate. Besides, we activate the learning rate restarting scheme whenever a new selection action is initiated. More experimental results and a brief analysis are provided below.

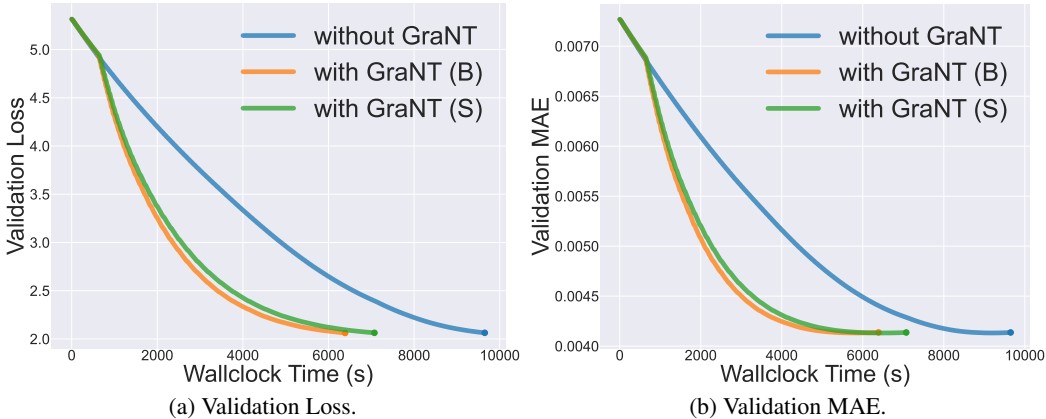

(a) Validation Loss.

(b) Validation MAE.

Figure 6: Validation set performance of graph-level tasks on QM9 (regression).

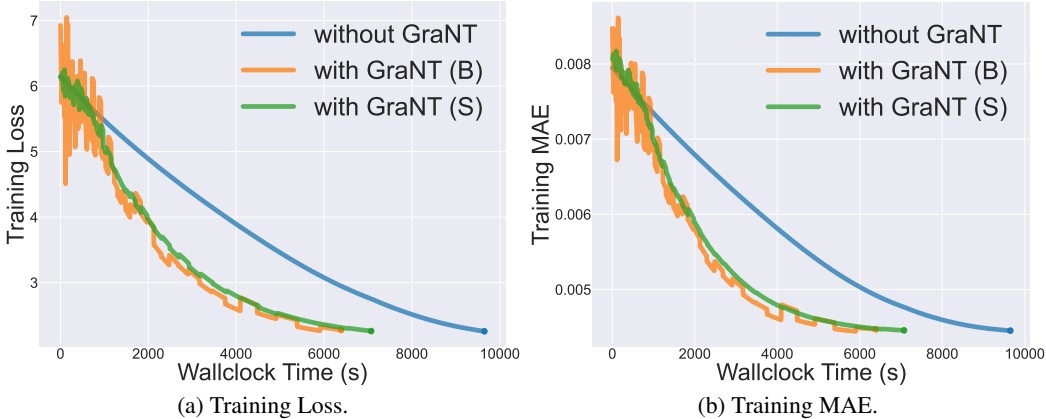

(a) Training Loss.

(b) Training MAE.

Figure 7: Training set performance of graph-level tasks on QM9 (regression).

First, we show the validation and training performance for the regression task on QM9 in Figure 6 and Figure 7, along with the training performance for the regression task on ZINC in Figure 8. For the classification tasks, the training performance on ogbg-molhiv is displayed in Figure 9. Moreover, the validation / training performance on ogbg-molpcba are provided in Figure 10 and Figure 11, respectively.

For QM9, as shown in Figure 6, GraNT (B) and GraNT (S) require less time than "without GraNT" and show a quicker decline in validation loss and MAE curves. In the early stages, the curves follow a nearly linear trend. This occurs because the learning rate is small, allowing the model to learn simple features and adjust its parameters gradually at the beginning. As the optimizer moves through the parameter space with steady steps, the model begins to capture more complex features, resulting in a shift to a nonlinear phase in the validation loss and MAE. In Figure 7, the training loss and MAE curves show significant fluctuations early on, which can be attributed to the frequent selection process. This requires the model to rapidly learn and adapt to new teaching graphs. Over time, the curves take on a step-like shape, suggesting that the selection process is stabilizing and the model has become adequately adaptive. Similarly, the training loss and MAE curves for ZINC in Figure 8 display similar trends.However, it is worth mentioning that GraNT (B) and GraNT (S) cut down the training time by around 3.5 hours for ZINC compared to "without GraNT", all while maintaining similar performance.

For ogbg-molhiv, as shown in Figure 9, the training ROC-AUC curve for GraNT (B) and GraNT (S) fluctuates due to the underlying data imbalance. However, the overall trend and final results align with expectations, even surpassing the "without GraNT" curve. To showcase the generalizability and scalability of GraNT, we further evaluate it on the large-scale multi-classification benchmark dataset ogbg-molpcba, which is 10 times larger than the ogbg-molhiv dataset. Figure 10 shows that the validation AP (Average Precision) curves for GraNT (B) and GraNT (S) consistently outperform the "without GraNT" curve in the mid-to-late stages. Additionally, it is notable that GraNT (B) and GraNT (S) reduce the training time by approximately **13.8** hours compared to the "without GraNT" setup. These results further emphasize GraNT's ability to

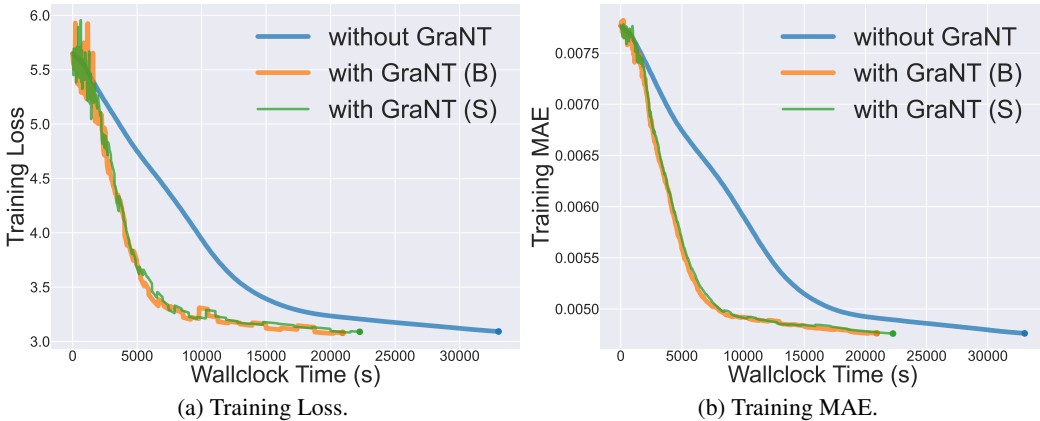

(a) Training Loss.

(b) Training MAE.

Figure 8: Training set performance of graph-level tasks on ZINC (regression).

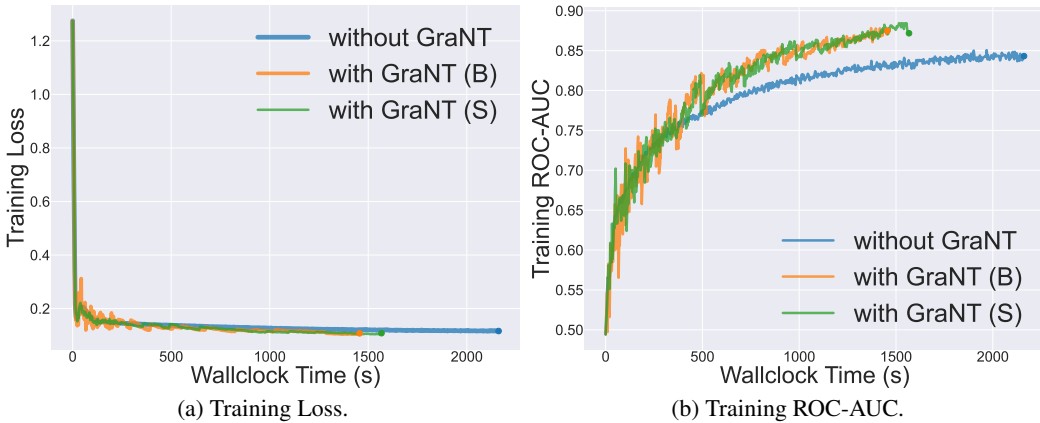

(a) Training Loss.

(b) Training ROC-AUC.

Figure 9: Training set performance of graph-level tasks on ogbg-molhiv (classification).

enhance training time efficiency on large-scale and complex datasets, particularly in the fields of chemistry and biomedical sciences. Additionally, the training loss and AP curves are shown in Figure 11.

Furthermore, on the AMD Instinct MI210 (64GB) device, we also validate the effectiveness of GraNT for graph-level tasks on the QM9 dataset, highlighting its cross-device effectiveness. As illustrated in Figure 12, GraNT (B) and GraNT (S) converge more quickly than "without GraNT," showing a faster decline in validation loss and MAE. Roughly speaking, GraNT (B) and GraNT (S) save more than 40% of the time compared to "without GraNT" while achieving comparable results.The training curves are displayed in Figure 13.

## C.3. Node-level Tasks

We train the GCN with GraNT (B), GraNT (S), and "without GraNT" using a standard experimental setup for node-level tasks, while applying the same *curriculum learning* strategy used in graph-level tasks. To evaluate GraNT on synthetic data, we utilize graphon (Xu et al., 2021; Xia et al., 2023) to create the gen-reg and gen-cls datasets. Specifically, we set the resolution of the synthetic graphon to 1000×1000, which produces graphs that can be easily aligned by arranging the node degrees in strictly increasing order. Additionally, each graph contains approximately 100 nodes, with a total of 50k graphs across both datasets. Lastly, we use a 2-layer GCN with a specific initialization to assign regression properties or classification labels to each node, while the dimension of node features in all graphs is set to 40.

The training performance results for gen-reg and gen-cls are shown in Figure 14 and Figure 15. In particular, Figure 14 clearly demonstrates that GraNT converges more quickly in terms of wallclock time compared to "without GraNT," which initially drops faster. This faster initial drop may occur because, for gen-reg, training on all the graphs provides sufficient information about the implicit mapping from graphs to their properties, resulting in better validation performance in the early

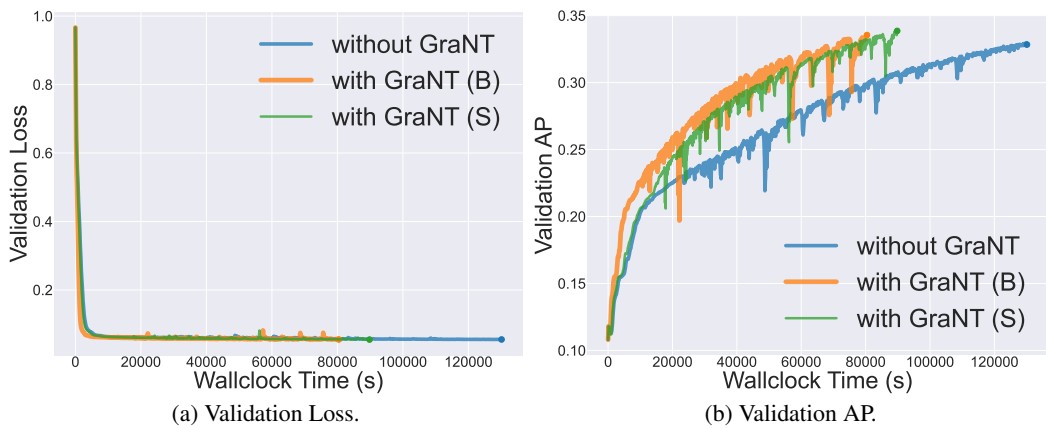

Figure 10: Validation set performance of graph-level tasks on ogbg-molpcba (multi-task classification).

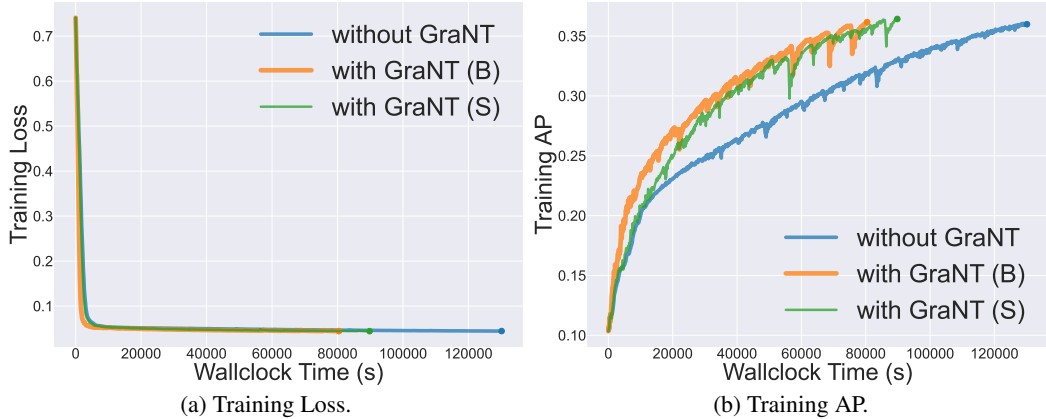

Figure 11: Training set performance of graph-level tasks on ogbg-molpcba (multi-task classification).

epochs, even though it requires more computational time. Over time, GraNT selects more representative teaching graphs (those with larger gradients), gradually accumulating enough information for the implicit mapping. In Figure 15, GraNT (B) takes the least time and shows a faster decline in the training loss curve compared to GraNT (S), while also reaching the final ROC-AUC results more quickly. Additionally, the training ROC-AUC curve exhibits significant oscillations early on, which can be attributed to the frequent selection of a diverse set of teaching graphs and the label imbalance.

In addition, we conduct experiments on node-level tasks with the gen-cls dataset using the AMD Instinct MI210 (64GB) device, further showcasing cross-device generalizability of GraNT. Specifically, GraNT (B) and GraNT (S) save about one-third of the time compared to without GraNT, while achieving even better performance. The validation and training performance results are shown in Figure 16 and Figure 17, respectively. As observed, when the loss curves flatten at their lowest points, the final validation and training ROC-AUC curves for GraNT (B) and GraNT (S) surpass those of the "without GraNT" baseline, highlighting GraNT's effectiveness in improving performance while reducing time consumption.

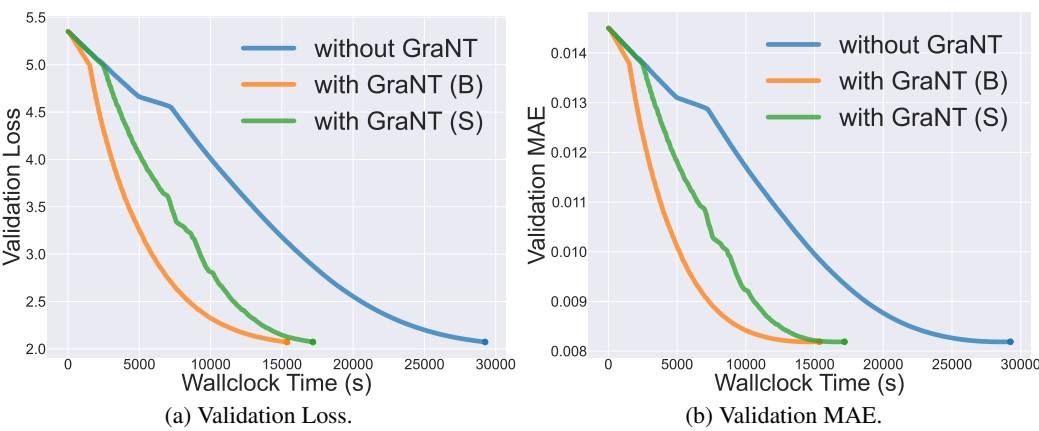

(a) Validation Loss.

(b) Validation MAE.

Figure 12: Validation set performance of graph-level tasks for QM9 (regression) on AMD device.

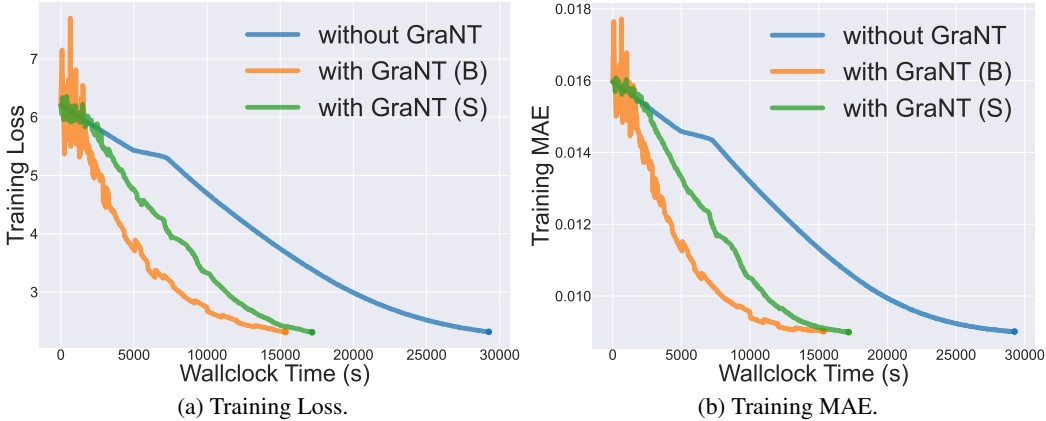

(a) Training Loss.

(b) Training MAE.

Figure 13: Training set performance of graph-level tasks for QM9 (regression) on AMD device.

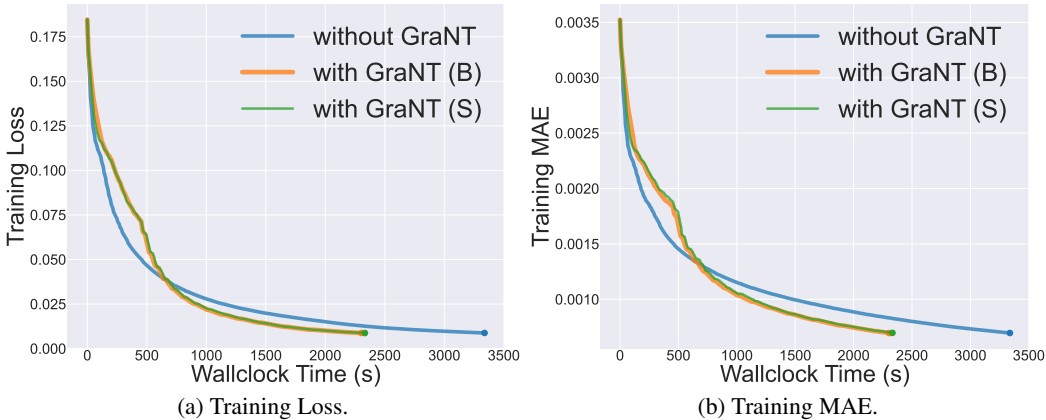

(a) Training Loss.

(b) Training MAE.

Figure 14: Training set performance of node-level tasks on gen-reg (regression).

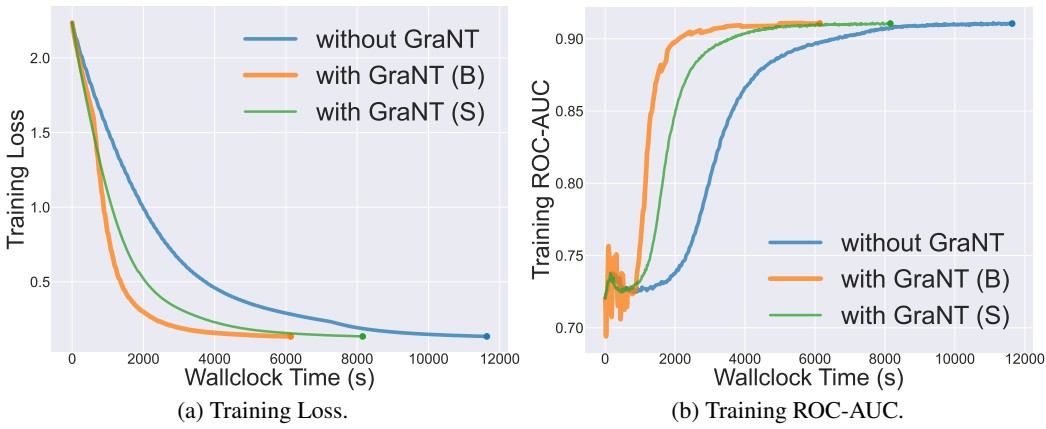

(a) Training Loss.

(b) Training ROC-AUC.

Figure 15: Training set performance of node-level tasks on gen-cls (classification).

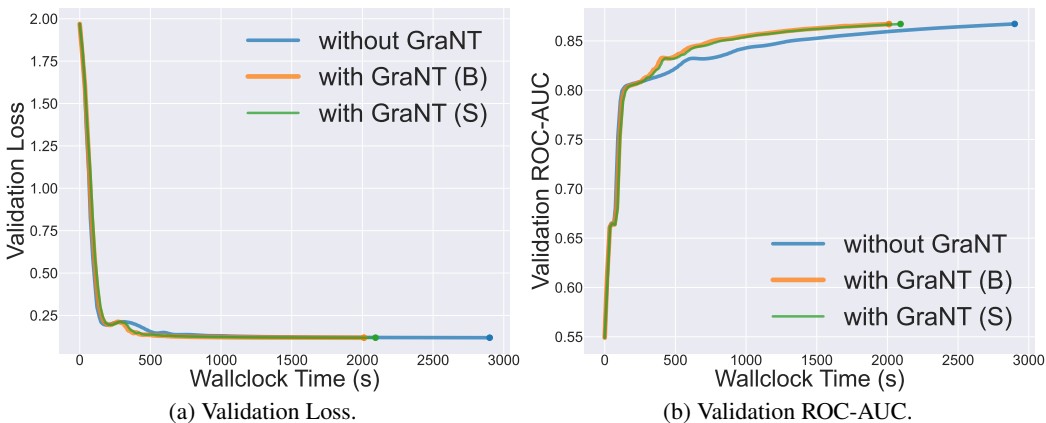

(a) Validation Loss.

(b) Validation ROC-AUC.

Figure 16: Validation set performance of node-level tasks for gen-cls (classification) on AMD device.

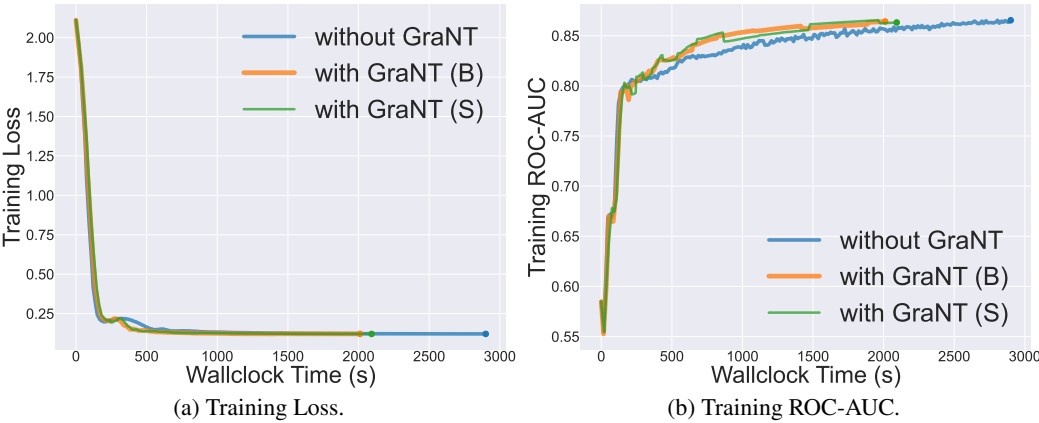

(a) Training Loss.

(b) Training ROC-AUC.

Figure 17: Training set performance of node-level tasks for gen-cls (classification) on AMD device.

