# OpenReview forum: "Nonparametric Teaching for Graph Property Learners"
_ICML.cc/2025/Conference — ICML 2025 spotlightposter_

### Official Review · Reviewer_2c6k · 2025-03-08

**Overall Recommendation:** 3

**Summary:**

The paper introduces GraNT, a novel paradigm that applies nonparametric teaching principles to accelerate the training of graph property learners (specifically GCN). By establishing a theoretical link between traditional parameter-based gradient descent and functional gradient descent, the authors design a greedy algorithm that selects a subset of graphs—those with the largest prediction discrepancies—to expedite convergence. Extensive experiments on both graph-level and node-level tasks across several benchmark datasets demonstrate that GraNT significantly reduces training time while preserving generalization performance.

**Claims And Evidence:**

Yes.

**Essential References Not Discussed:**

One major issue of this paper is that while the author claims to solve the efficiency, non of the efficient GNN papers are discussed nor compared.

**Experimental Designs Or Analyses:**

The current evaluation compares only different variants of GraNT. Including comparisons with established efficient GNN models and state-of-the-art GNNs would better contextualize the contributions and highlight the advantages of the proposed method.

**Methods And Evaluation Criteria:**

Yes. The paper effectively bridges nonparametric teaching with GCN training, offering a fresh perspective by interpreting the evolution of GCNs through the lens of functional gradient descent.

**Other Comments Or Suggestions:**

NA

**Other Strengths And Weaknesses:**

Although the paper claims that training GCNs is costly, it would be more compelling if it detailed how current efficient graph libraries (e.g., DGL, PyG) fall short in addressing these challenges. A more comprehensive discussion of experimental settings and implementation specifics would clarify how GraNT overcomes these limitations.

Besides, the code and data are not available.

**Questions For Authors:**

NA

**Relation To Broader Scientific Literature:**

NA

**Theoretical Claims:**

The overall proof seems correct. Yet, I didn't go through the details.

The issue is that the proofs are primarily based on (two-layer) GCNs, a special case of a relatively old baseline. Extending the theoretical framework and empirical validation to more recent architectures (e.g., Graph Attention Networks) would strengthen the claim of general applicability and broaden the impact of the work.

---

> ### Author Rebuttal · Authors · 2025-03-30
>
> Thanks for many constructive comments. We are deeply appreciative of the reviewer’s efforts to help us improve our paper. We take all comments seriously and try our best to address every raised concern. We sincerely hope that our response can resolve your concerns. Any follow-up questions are welcome.
>
> **[Theoretical Claims]** Sorry for the confusion. We would like to clarify that the theoretical analysis does not rely on the GCN being limited to two layers; it can be of any depth. Additionally, our empirical validation includes experiments with different depths and widths, as shown in Table 4. We agree with the reviewer that expanding the theoretical framework and empirical validation to more recent architectures would enhance the broader applicability of the work. We also believe that the concept and analysis behind GraNT have significant potential for wider extensions. Analyzing GCNs is an important foundational step in applying nonparametric teaching for graph property learning, which paves the way for further advancements. However, this is beyond the scope of the current paper and will be explored in future research. In response to your inquiry, we have applied the GraNT algorithm to GIN [1] on the ogbg-molhiv dataset and will provide further discussion on this in the revised version.
>
> |   |   GIN [1] w/o GraNT    |   GIN [1] w/ GraNT (B)   | GIN [1] w/ GraNT (B)   |
> |---|--------------|-----------|--------------|
> | Training Time |  1446.91  |  982.66 (-32.09%)   |     965.37 (-33.28%)  |
> |   ROC-AUC  | 0.7694   |  0.7714  |     0.7705 |
>
> **[Experimental Designs Or Analyses]** Thank you for your constructive feedback. We present comparisons of GraNT with recent works [2-6] below. Our results show that, with theoretical justification, GraNT outperforms these methods in terms of both training time and specific metrics. We will provide further discussion on this in the revised version to better emphasize the context and contributions of our work.
>
> **ogbg-molhiv**
> |  | GMoE-GCN [2] | GMoE-GIN [2] | GDeR-GCN$^\alpha$[3] | GDeR-PNA$^\alpha $ [3] | GCN [4] | GCN+virtual node[4] | GraNT (B)  | GraNT (S)  |
> |---|------|----|----|----|----------|------|-------|------------|
> | Training Time |  3970.16  |  3932.06   |   1772.23   |   5088.88   | 2888.80 | 3083.16 | 1457.39 | 1597.69 |
> | ROC-AUC  |  0.7536  |   0.7468   |    0.7261   |  0.7616  | 0.7385 | 0.7608 | 0.7676  |  0.7705 |
>
> $^\alpha$: batch_size=500, retain_ratio=0.7.
>
> **QM9**
> |   | MOAT$^\gamma$  [5]   |  AL-3DGraph$^\delta$ [6] |  AL-3DGraph$^\zeta$ [6] |  AL-3DGraph$^\tau$ [6]   |  GraNT (B)   | GraNT (S)   |
> |---|------|--------|-----|------|-------|-------|
> | Training Time |  69000  |     9200.27 | 9364.74 | 12601.77  |     6392.26  |  7076.37 |
> |   MAE      |  0.0236  |     0.7991 | 0.4719  | 0.1682  |    0.0051 | 0.0051 |
>
> $^\gamma$: The training time is calculated as time_per_epoch * epochs = 92 * 750. The MAE value reported is from the paper.
>
> $^\delta$: lr=5e-5, batch_size = 256, which matches our settings.
>
> $^\zeta$: lr=5e-4, batch_size = 256.
>
> $^\tau$: lr=5e-4, batch_size = 32, which corresponds to the default settings used in the provided code for that paper.
>
> [1] How powerful are graph neural networks?. – ICLR’19.
>
> [2] Graph mixture of experts: Learning on large-scale graphs with explicit diversity modeling. – NeurIPS’23.
>
> [3] GDeR: Safeguarding efficiency, balancing, and robustness via prototypical graph pruning. – NeurIPS’24.
>
> [4] Semi-supervised classification with graph convolutional networks. – ICLR’17.
>
> [5] Graph Prompting for 3D Molecular Graphs. - CIKM’24.
>
> [6] Empowering Active Learning for 3D Molecular Graphs with Geometric Graph Isomorphism. – NeurIPS’24.
>
> **[Essential References Not Discussed]** Sorry for the confusion. We would like to clarify that the discussion on learning efficiency (Chen et al., 2018; Liu et al., 2022b; Zhang et al., 2023c) is included in the Related Works section, covering topics like normalization (Cai et al., 2021), graph decomposition (Xue et al., 2023), and lazy updates (Narayanan et al., 2022). We will enhance the presentation and include more detailed discussion in the revised version.
>
> **[W1]** Thank you for your suggestions. We would like to clarify that this paper is algorithmic in nature, introducing GraNT, a novel paradigm that improves the learning efficiency of graph property learners (GCNs) through nonparametric teaching. Our implementation leverages the PyG graph library, which is designed to optimize compilation efficiency, a concern that is independent of the main focus of our work.
>
> **[W2]** We have included the pseudocode in Algorithm 1, which outlines the core idea of our implementation, and our experiments are based on this. Detailed experimental settings and implementation specifics can be found in Appendix C. The code and data will be made available.

---

> > ### Comment · Reviewer_2c6k · 2025-04-05
> >
> > The authors did a great job in addressing my concerns. I sincerely appreciate their effort for adding new experiments. Since my concerns are addressed. I've raised my score accordingly.

---

> > > ### Author Response · Authors · 2025-04-05
> > >
> > > Thank you for your constructive feedback. We're pleased that our response has addressed your concerns. Should you have any follow-up questions, please feel free to reach out. We will certainly incorporate any new and useful results in the revision. We also appreciate your updated score.

---

### Official Review · Reviewer_7pFN · 2025-03-10

**Overall Recommendation:** 4

**Summary:**

This paper presents GraNT (Graph Nonparametric Teaching), a novel framework that improves the learning efficiency of graph property learners (GCNs) using nonparametric teaching.

**Claims And Evidence:**

Yes

**Essential References Not Discussed:**

No

**Experimental Designs Or Analyses:**

Yes

**Methods And Evaluation Criteria:**

Yes

**Other Comments Or Suggestions:**

N/A

**Other Strengths And Weaknesses:**

1. Long training times due to slow convergence are common in many machine learning tasks, particularly in AI4Science applications, as explored in this paper. Therefore, I believe the proposed approach's improvement in learning efficiency is highly valuable.

2. Section 4 is well-structured, progressing logically from theoretical analysis to algorithm design, making the presentation clear and effective.

3. Graph Convolutional Networks (GCNs) are a fundamental GNN architecture, but more advanced GNNs have been developed with improved expressivity, scalability, and the ability to capture complex graph structures. Examples include MPNN, GAT, GIN, Graph Transformer Networks, as well as high-order and equivariant GNNs. These architectures are now more commonly used for complex regression and classification tasks at both graph and node levels, often requiring even longer training times. I’m curious whether the proposed approach can also be applied to these more advanced GNNs beyond GCN.

**Questions For Authors:**

N/A

**Relation To Broader Scientific Literature:**

Not sure if this proposed approach can be applied to more GNNs beyond GCN.

**Theoretical Claims:**

Yes

---

> ### Author Rebuttal · Authors · 2025-03-30
>
> Thanks for the encouraging comments and constructive suggestions. We sincerely thank the reviewer's efforts for helping us improve the paper. We hope that our response resolves your concerns.
>
> **[W3]** Very thoughtful question! We believe the idea and analysis behind GraNT have significant potential for broader extensions. As a fundamental neural network architecture for graph property learning, analyzing GCNs is an essential step in applying nonparametric teaching within this context before exploring further extensions. As mentioned in the Concluding Remarks and Future Work section, exploring other GNNs beyond GCN could be an intriguing direction. This could involve examining the core mechanisms of specific GNNs through the GNTK theory. Some relevant works on GNTK that might be insightful are [1,2]. However, this is outside the scope of the current paper and will be addressed in future work. In response to your curiosity, we have applied the GraNT algorithm to GIN [3] on the ogbg-molhiv dataset and will include additional discussion on this in the revised version.
>
> |   |   GIN [3] w/o GraNT    |   GIN [3] w/ GraNT (B)   | GIN [3] w/ GraNT (B)   |
> |---|--------------|-----------|--------------|
> | Training Time |  1446.91  |  982.66 (-32.09%)   |     965.37 (-33.28%)  |
> |   ROC-AUC  | 0.7694   |  0.7714  |     0.7705 |
>
> [1] Graph neural tangent kernel: Convergence on large graphs. – ICML’23.
>
> [2] Graph neural tangent kernel: Fusing graph neural networks with graph kernels. -NeurIPS’19.
>
> [3] How powerful are graph neural networks?. – ICLR’19.

---

### Official Review · Reviewer_WzeW · 2025-03-12

**Overall Recommendation:** 3

**Summary:**

In this paper, the authors innovatively introduce a training paradigm termed Graph Nonparametric Teaching (GraNT) designed for graph property learners. Their main idea is to reinterpret the training of GCNs through the lens of nonparametric teaching, which selects training examples (graphs) strategically to accelerate learning. The authors demonstrate that parameter-based gradient descent updates in GCNs align with functional gradient descent in nonparametric teaching. Experiments show that the proposed approach can reduce training time to some extent across various graph-level and node-level tasks (regression and classification) without sacrificing much generalization performance.

**Claims And Evidence:**

The main claims are supported by both analytical derivations and extensive experimental validation across multiple datasets (e.g., QM9, ZINC, ogbg-molhiv, ogbg-molpcba).

**Essential References Not Discussed:**

NA.

**Experimental Designs Or Analyses:**

The experiment setup makes sense, but is not sufficient. I indeed have some concerns regarding the experiment as what will be elaborated below.

**Methods And Evaluation Criteria:**

The method is logically sound. The evaluation criteria make sense to me.

**Other Comments Or Suggestions:**

Minor typo:
- Page 6, Line 295: “GNKT” should be corrected to “GNTK”.

**Other Strengths And Weaknesses:**

**Strengths:**

- The theoretical integration of nonparametric teaching into graph property learning is original.

- Experiments are performed on some large-scale graph benchmarks.

- The presentation is clear and well-written.

- Detailed derivations and additional results are provided in the appendix.

**Weaknesses:**

I do have some major concerns regarding the experiments of this paper.

First of all, the comparative experiments are far from sufficient. The authors are suggested to compare GraNT with more GNNs. Also, it is known that the results on some datasets, such as ZINC, are sometimes unstable (see the benchmark paper [a]). It is suggested that the std be reported for the related results. This is critical because it can demonstrate whether the performance of GraNT is consistently better, especially given the method's reliance on gradient magnitude for teaching graph selection, which might be sensitive to label distribution imbalance and impact stability.

[a] Benchmarking Graph Neural Networks.

Furthermore, there is limited exploration of the algorithm's sensitivity to hyperparameters, such as the number of selected graphs in GraNT.

**Questions For Authors:**

Please address my aforementioned major concerns on paper experiments, such as insufficient comparative experiments and lack of stability analysis.

**Relation To Broader Scientific Literature:**

The paper is well-positioned within the literature of graph neural networks.

**Theoretical Claims:**

I didn't check completely the proof in the appendix. But it appears to me that the proof is well-presented.

---

> ### Author Rebuttal · Authors · 2025-03-30
>
> Thanks for the useful comments. We are deeply appreciative of the reviewer’s efforts to improve our paper. We take all comments seriously and try our best to address every raised concern. We sincerely hope that our response resolves your concerns.
>
> **[W1]** Thank you for your helpful feedback in improving our paper. Below, we present a comparison of GraNT with the recent works GMoE [1] and GDeR [2]. Our results demonstrate that GraNT outperforms these approaches in both training time and specific metrics.
>
> We will include a more detailed discussion of this comparison in the revision.
>
> **ogbg-molhiv**
> |  | GMoE-GCN [1] | GMoE-GIN [1] | GDeR-GCN$^\alpha$[2] | GDeR-PNA$^\alpha$ [2] | GCN [3] | GCN+virtual node[3] | GraNT (B)  | GraNT (S)  |
> |---|------|----|----|----|----------|------|-------|------------|
> | Training Time |  3970.16  |  3932.06   |   1772.23   |   5088.88   | 2888.80 | 3083.16 | 1457.39 | 1597.69 |
> | ROC-AUC  |  0.7536  |   0.7468   |    0.7261   |  0.7616  | 0.7385 | 0.7608 | 0.7676  |  0.7705 |
>
> $^\alpha$: batch_size=500, retain_ratio=0.7.
>
> **QM9**
> |   | MOAT$^\gamma$  [4]   |  AL-3DGraph$^\delta$ [5] |  AL-3DGraph$^\zeta$ [5] |  AL-3DGraph$^\tau$ [5]   |  GraNT (B)   | GraNT (S)   |
> |---|------|--------|-----|------|-------|-------|
> | Training Time |  69000  |     9200.27 | 9364.74 | 12601.77  |     6392.26  |  7076.37 |
> |   MAE      |  0.0236  |     0.7991 | 0.4719  | 0.1682  |    0.0051 | 0.0051 |
>
> $^\gamma$: The training time is calculated as time_per_epoch * epochs = 92 * 750. The MAE value reported is from the paper.
>
> $^\delta$: lr=5e-5, batch_size = 256, which matches our settings.
>
> $^\zeta$: lr=5e-4, batch_size = 256.
>
> $^\tau$: lr=5e-4, batch_size = 32, which corresponds to the default settings used in the provided code for that paper.
>
> [1] Graph mixture of experts: Learning on large-scale graphs with explicit diversity modeling. – NeurIPS’23.
>
> [2] GDeR: Safeguarding efficiency, balancing, and robustness via prototypical graph pruning. – NeurIPS’24.
>
> [3] Semi-supervised classification with graph convolutional networks. – ICLR’17.
>
> [4] Graph Prompting for 3D Molecular Graphs. - CIKM’24.
>
> [5] Empowering Active Learning for 3D Molecular Graphs with Geometric Graph Isomorphism. – NeurIPS’24.
>
> **[W2]** Thanks for your help in improving our paper. We have included the results with standard deviations for all validations. This demonstrates that the proposed GraNT not only achieves training time efficiency with theoretical justification but also exhibits stability. We will fix it.
>
> | GraNT | Dataset         | MAE ↓      | ROC-AUC ↑       | AP ↑       |
> |-------|-----------------|------------|-----------------|------------|
> | ✗     | QM9            | 0.0051$\pm$0.0009 | -               | -          |
> | ✗     | ZINC           | 0.0048$\pm$0.0004   | -              | -          |
> | ✗     | ogbg-molhiv    | -          | 0.7572$\pm$0.0005       | -          |
> | ✗     | ogbg-molpcba   | -          | -               | 0.2579$\pm$0.0058     |
> | ✗     | gen-reg        | 0.0007$\pm$0.0001     | -               | -          |
> | ✗     | gen-cls        | -          | 0.9150$\pm$0.0024        | -          |
> | ✓ (B) | QM9            | 0.0051$\pm$0.0009     | -               | -          |
> | ✓ (B) | ZINC           | 0.0048$\pm$0.0004     | -               | -          |
> | ✓ (B) | ogbg-molhiv    | -| 0.7676$\pm$0.0036          | -          |
> | ✓ (B) | ogbg-molpcba   | -| -               | 0.2580$\pm$0.0026 |
> | ✓ (B) | gen-reg        | 0.0007$\pm$0.0001     | -               | -          |
> | ✓ (B) | gen-cls        | -          | 0.9157$\pm$0.0013      | -          |
> | ✓ (S) | QM9            | 0.0051$\pm$0.0009     | -               | -          |
> | ✓ (S) | ZINC           | 0.0048$\pm$0.0004     | -               | -          |
> | ✓ (S) | ogbg-molhiv    | -| 0.7705$\pm$0.0027      | -          |
> | ✓ (S) | ogbg-molpcba   | -| -               | 0.2579$\pm$0.0047     |
> | ✓ (S) | gen-reg        | 0.0007$\pm$0.0001    | -               | -          |
> | ✓ (S) | gen-cls        | -          | 0.9157$\pm$0.0014      | -          |
>
> **[W3]** Thanks for your help in improving our paper. The analysis of the hyperparameter, start ratio, under GraNT(B) is outlined below:
>
> We will include a more detailed discussion in the revision.
>
> |Dataset| Start_Ratio | 0.05 | 0.1  | 0.2  | 0.4  | 0.8  | full |
> |---|-------|------|-------|-------|-------|-------|-------|
> | QM9 | MAE   | 0.0051 | 0.0053 | 0.0053 | 0.0053 | 0.0053 | 0.0051 |
> | QM9 | Training time (s) | 6392.26 | 6974.30 | 7918.51 | 10828.18 | 14081.66| 9654.81 |
> | ogbg-molhiv | ROC-AUC | 0.7546 | 0.7676 | 0.7652 | 0.7618| 0.7592 | 0.7572|
> | ogbg-molhiv | Training time (s) | 1362.09 | 1457.39 | 1719.43 | 2157.05 | 3173.92| 2163.5 |
> | gen-cls | ROC-AUC | 0. 9157| 0.9156 | 0. 9156| 0.9156| 0.9156| 0.9150|
> | gen-cls | Training time (s) | 6145.72 | 6237.92 | 6939.95 | 9459.22 | 13153.81 | 11662.25 |
>
> **[Typo]** Thanks for pointing it out. We will fix it in the revision.

---

### Decision · Program_Chairs · 2025-05-01

**Decision:**

Accept (spotlight poster)

**Comment:**

The paper introduces graph nonparametric teaching to address the slow learning of standard graph property learners. Specifically, the approach introduces a theoretical framework for implicit mappings of properties via example selection. Theoretical and empirical results show the efficacy of the approach as compared to standard graph property learning approaches. The reviewers all agree that the paper is interesting, makes a novel contribution, and opens new directions for further research.